# SOMA: Efficient Multi-turn LLM Serving via Small Language Model

## Abstract

Large Language Models (LLMs) are increasingly deployed in multi-turn dialogue settings where preserving conversational context across turns is essential. A standard serving practice concatenates the full dialogue history at every turn, which reliably maintains coherence but incurs substantial cost in latency, memory, and API expenditure, especially when queries are routed to large proprietary models. Existing approaches often struggle to balance the trade-off between response quality and efficiency. We propose a framework that exploits the early turns of a session to estimate a local response manifold and then adapt a smaller surrogate model to this local region for the remainder of the conversation. Concretely, we learn soft prompts that maximize semantic divergence between the large and surrogate small language models' responses to surface least-aligned local directions, stabilize training with anti-degeneration control, and distill the mined cases into localized LoRA fine-tuning so the surrogate runs without prompts at inference. A simple gate enables a one-time switch with rollback on drift. We further provide a theoretical analysis for key components in SOMA. Extensive experiments show the effectiveness of SOMA. The source code is provided at:
`https://anonymous.4open.science/r/SOMA-D377`.

## 1 Introduction

Large language models (LLMs) such as the GPT series (Radford et al., 2019; Brown et al., 2020; Achiam et al., 2023), LLaMA (Touvron et al., 2023), Claude (Anthropic, 2023), and DeepSeek (Guo et al., 2025) have demonstrated strong performance in real-world machine-learning-as-a-service (MLaaS) applications, ranging from chat assistants to code generation (Park & Kulkarni, 2023; Dong et al., 2023; Liu et al., 2024). As LLMs are increasingly deployed in interactive settings, *multi-turn LLM serving*, involving extended interactions between humans and LLMs or among multiple LLM agents, has emerged as a key research focus, as it better reflects real-world usage scenarios (Yi et al., 2024; Li et al., 2025). Existing research reveals that multi-turn interactions are widespread, underscoring the need for serving systems capable of handling extended conversations in a context-aware manner (Chen et al., 2024a; Gao et al., 2024). However, supporting efficient context-dependent multi-turn interaction remains a key challenge, as most LLM serving systems are stateless and require resending the entire conversation history, including all prior queries and responses, with each new query to generate a new response (Ananda, 2025; Moon, 2025). This leads to redundant computation, high latency, and rising serving costs as conversations lengthen.

Previous work has explored efficient multi-turn LLM serving through two main approaches. One line of work focuses on *single-model methods* that compress dialogue history (Wang et al., 2025; Chen et al., 2024b; Xiao et al., 2024), retrieve memory from external modules (Melz, 2023; Gutiérrez et al., 2024), or reuse attention computations (Gao et al., 2024; Jeong & Ahn, 2025; Anthropic, 2024). However, these still rely heavily on large LLMs for every turn, leading to high monetary cost, latency, and GPU usage. They also often truncate or overlook extended context, limiting reasoning over long dialogues. Another line adopts *multi-model methods*, routing simple queries to smaller models while escalating harder ones to larger LLMs (Behera et al., 2025; Schick et al., 2023; Ding et al., 2024; Shnitzer et al., 2023). Yet, small models struggle to generalize across dialogue complexity, and model switching introduces additional overhead. Moreover, LLMs are known to over-rely on early turns (Xiao et al., 2023; Laban et al., 2025), compounding the difficulty of maintaining coherence in multi-turn settings.

***Is it possible to achieve an efficient, context-aware multi-turn LLM serving framework that avoids recomputing the full history at every turn while preserving response quality?***

To achieve this goal, we perform in-depth explorations of real-world multi-turn dialogues across different domains and reveal an interesting long-tail distribution in token counts across turns: early turns are substantially longer, while later turns gradually decrease in length. This phenomenon aligns with the intuition of prior work that early turns typically carry the substantive openings set issues and anchors, including questions, requests, and proposals, while later turns are typically minimal acknowledgments (He et al., 2018; Stolcke et al., 2000). This long-tail trend gives rise to an intuitive idea: since later turns are relatively short, a smaller language model might suffice to generate responses more cost-efficiently. However, the bottleneck lies in the "big head": if the small model must still process the early, information-dense context, the response quality will be degraded. Specifically, while a small language model may behave reasonably at the start, its responses drift from the large model as the dialogue progresses because most grounding is established early, and later turns remain highly dependent on that context. Therefore, simply handing later turns to a small model without modeling the accumulated context degrades quality. To address this, the small language model must not only process shorter inputs but also approximate the larger model's behavior within the local manifold of its output or hidden space to capture the contextual dependencies shaped by prior dialogue. This defines a local manifold approximation problem, where the small language model aims to replicate the target model's behavior induced by the current conversational context.

Building on these insights, we propose a novel framework for efficient multi-turn LLM serving that enables a small language model to locally approximate the behavior of a large language model within a constrained region of the reasoning manifold. Specifically, we present SOMA (*S*oft-prompts for l*O*cal *M*anifold *A*pproximation) to dynamically adapt the small language model to the local behavior of the larger model conditioned on early turn interactions. This is achieved through a three-stage pipeline: (1) *Soft prompt tuning*, where we efficiently explore the local reasoning manifold induced by the early conversational context to identify directions of maximal behavioral divergence between the small and large language model ; (2) *Localized fine-tuning*, where we efficiently fine-tune the small language model on a small number of input–output pairs to align it with the larger model within this context-specific region of the manifold; and (3) *Efficiency inference*, where we incorporate the extractive summary to minimize computational overhead and the rollback mechanism that monitor potential topic shift to maintain service quality. Together, these components allow the small model to effectively approximate the larger model's reasoning process within the context of a given session, enabling both cost-effective and context-aware multi-turn serving. Extensive experiments on real-world datasets show the effectiveness of our proposed method. Our contributions are:

- **Long-tail pattern in multi-turn dialogues.** We first reveal a previously under-explored long-tail pattern in multi-turn dialogues: the first few turns concentrate heavy context, while later turns are shorter yet more dependent on previous turns. This key empirical characterization suggests that substantial computational and monetary savings can be achieved if a smaller and cheaper language model can replace a large one to process the later turns when given the accumulated context.

- **SOMA: efficient multi-turn serving.** It first learns soft prompts that expose the largest surrogate–original response dissimilarity, then adapts the surrogate localized LoRA accordingly, enabling prompt-free inference with a simple cosine gate for switching and rollback.

- **Theory analysis and empirical evaluation.** We provide concentration-based bounds for switching, coverage guarantees for prompt-direction search, and suboptimality limits for selected directions. Guided by these results, empirical studies show the effectiveness of SOMA in real world.

## 2 PRELIMINARIES

### 2.1 NOTATIONS

In this paper, a multi-turn dialogue prefix of length $k$ is $\mathcal{D}_k = \{(q_1, a_1), \ldots, (q_k, a_k)\}$ where $q_t$ is the user query at turn $t$, and $a_t$ is the corresponding model response. $F$ represents the original proprietary black-box LLM, which is a common setting in machine-learning-as-a-service (MLaaS), and $G$ is the surrogate small language model. The textual response at turn $t$ is $a_t^M$ for $M \in \{F, G\}$. Let $f_M(\cdot)$ be a feature map to the hidden space, and $\mathbf{h}_t = f_M(q_{\leq t}) \in \mathbb{R}^d$ the hidden state at turn

$t$. The first $k$ hidden states form $\mathcal{H}_k = \{\mathbf{h}_1, \ldots, \mathbf{h}_k\}$ and induce a local manifold $\mathcal{M}_k^M \subset \mathbb{R}^d$. A length-$L$ soft prompt is $\mathbf{P} \in \mathbb{R}^{L \times d}$. More details are given in Appendix A.

## 2.2 Exploring The Token-Turn Patterns in Multi-Turn LLM Dialogues

While LLMs process the context from all previous turns equally by default, not all turns may contribute equally to the dialogue's contextual demands. Identifying when the model truly needs to rely on long-range context can inform adaptive strategies. Therefore, we begin with an empirical investigation of how information and contextual complexity are distributed across dialogue turns in real-world settings. To broadly cover real-world multi-turn dialogue, we select four complementary settings spanning different domains and dialogue purposes: *ShareGPT* (Chen et al., 2024a): open-ended, dyadic human–LLM chat; *ReMeDi* (Yan et al., 2022): task-oriented, asymmetric doctor–patient consultations; *Craigslist Bargain* (He et al., 2018): symmetric, goal-driven negotiation; and *Multi-Character* (agentlans, 2024): multi-party, role-play coordination. We then compute the average number of tokens usage at turn t over these dialogues and divide by the average token count at turn 1 in the

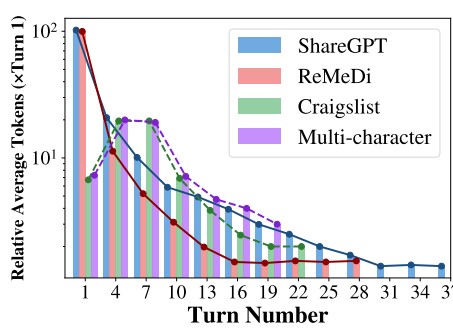

Figure 1: Relative average token count per turn (normalized by Turn 1) across four multi-turn dialogue datasets. All datasets show a long-tail pattern, where token usage drops sharply after the early turns and stabilizes in later turns.

same dialogue, yielding the "relative average tokens (× Turn 1)" curve. Building on this setup, our empirical analysis reveals a surprising *long-tail token distribution pattern* in multi-turn dialogues. As shown in Figure 1, the average tokens per turn drop steeply after the first few exchanges and then plateau at a much lower level across all four datasets. This finding aligns with prior observations that early turns typically carry the substantive openings set issues and anchors, including questions, requests, and proposals, while later turns are typically minimal acknowledgments (He et al., 2018; Stolcke et al., 2000). To the best of our knowledge, this is the first characterization of such long-tail distributions in multi-turn dialogues across the multi-tun dialogues in different domains. This phenomenon offers us important implications that current LLMs recompute full-context responses, even for the later lightweight turns, would result in large computational overhead.

## 2.3 Local Manifold Approximation For Multi-turn Dialogues

Building on the above analysis, we are motivated by an intuitive rationale: since later turns appear lightweight, a smaller and cheaper surrogate model might be sufficient to generate responses for them. However, this task is non-trivial, as empirical studies have shown that replacing the original large model with a smaller surrogate directly may lead to increasing discrepancies in performance, coherence, and contextual alignment (Chen et al., 2023; Koudounas et al., 2025). To better understand this limitation and guide a more effective solution, we first present the fundamental analysis of multi-turn LLM dynamics from the manifold perspective (Chui et al., 1994; Bengio et al., 2013), a view that has been widely adopted in computer vision, speech, and natural language processing (Fang et al., 2022; Turaga et al., 2008; Minh & Tuan, 2022). In the context of large language models, the manifold refers to a structured, lower-dimensional subspace within the model's high-dimensional embedding space where semantically meaningful internal representations are concentrated. These representations are typically captured by token-level or sequence-level embeddings extracted from the final layers of a transformer. In a multi-turn dialogue, the input at each turn consists of a token sequence formed by concatenating the prior dialogue history with the current user query. When this input is processed by the model, it is transformed into a high-dimensional embedding that corresponds to a specific point on the manifold. The model then decodes from this point to generate the output response, thereby reflecting its current understanding of the dialogue context. Details of the mathematical concepts are in Appendix C.

**Definition 2.1** (Manifold Hypothesis in LLMs). *A manifold $\mathcal{M} \subset \mathbb{R}^D$ refers to a lower-dimensional region of the contextual embedding space formed by the outputs of intermediate transformer layers, where semantically meaningful representations reside. Here, $D$ denotes the embedding dimension-*

*ality (e.g., hidden size of the LLM), and $n \ll D$ represents the intrinsic dimension of the representations induced by natural language inputs during multi-turn inference.*

During multi-turn interactions, each successive query typically introduces only minor variations, resulting in small perturbations to the input sequence. These perturbations lead to nearby shifts in the model's internal representation, as the hidden states of LLMs evolve smoothly with respect to small changes in input (Marro et al., 2025), and even minor variations between turns lead to nearby shifts in the model's representation space (Fu & Lapata, 2022). Importantly, these transitions do not occur arbitrarily in high-dimensional space but tend to follow a structured, low-dimensional manifold that captures the evolving semantic and contextual state of the conversation (Zhang & Dong, 2025; Dong et al., 2025). This perspective suggests that multi-turn dialogue progresses within a localized region of the model's manifold, and that maintaining coherence across turns depends on preserving alignment within this region. However, when the small surrogate model is directly fed the context at a later turn without adaptation, it often fails to track the localized progression established by the original model in earlier interactions. As a result, even small perturbations can shift the surrogate's internal representation away from the appropriate region of the manifold, leading to output responses that deviate from those of the original model and ultimately disrupt conversational continuity.

Based on the above analysis, to enable a surrogate model to behave similarly to the original model that has already processed the initial turns of dialogues, the challenge lies in ensuring that the surrogate produces contextually aligned responses without having access to the full capacity of the original model. Inspired by prior work on local manifold approximation (Chui et al., 1994; Karygianni & Frossard, 2014; Li & Dunson, 2020; Sober & Levin, 2020), we propose to approximate the local landscape of the original model's representation manifold activated by the dialogue prefix. If this local landscape is faithfully captured, the surrogate model can decode future responses that remain consistent with the behavior of the original model. This enables efficient multi-turn serving by replacing expensive original-model inference with lightweight generation guided by a locally approximated manifold. Formally, we frame this intuition as the problem below:

**Problem 1** (Local Manifold Approximation for Multi-Turn Interactions). *Given a dialogue prefix $\mathcal{D}_k = \{(q_1, a_1), \ldots, (q_k, a_k)\}$ and a black-box original model $F$, learn a surrogate $G$ whose local manifold matches that of $F$ under the same prefix. Formally, over a class of surrogates $\mathcal{G}$ we solve*

$$\min_{G \in \mathcal{G}} \ \mathrm{dist}\big(\mathcal{M}_k^G(\mathcal{D}_k), \ \mathcal{M}_k^F(\mathcal{D}_k)\big),$$

*where $\mathrm{dist}(\cdot, \cdot)$ is a metric to measure the manifold discrepancy, such as geodesic distance, average principal angle, or maximum mean discrepancy between subspaces.*

## 3   SOMA: LOCAL MANIFOLD APPROXIMATION BASED ON SOFT PROMPTS

In practice, directly accessing or comparing the latent manifolds $\mathcal{M}_k^G$ and $\mathcal{M}_k^F$ is intractable. Moreover, directly updating the surrogate's full parameter set to approximate the local manifold of the original model is computationally expensive and ineffective. One alternative is to fine-tune the surrogate on queries that yield the most dissimilar responses from the two models, as these queries would point to the small local changes where the two local manifolds are least aligned. However, a single dialogue provides only a few queries, which are insufficient to explore the local manifolds or reveal the least aligned small changes. To address this challenge, we therefore propose a novel framework SOMA (*S*oft-prompts for l*O*cal *M*anifold *A*pproximation). Given a multi-turn dialogue, in the first few turns, we perform lightweight soft-prompt tuning on the surrogate to learn soft prompts that, when concatenated with the queries, steer the interaction toward directions with the largest response differences from the original. Here, a 'direction' refers to a small additive perturbation in the surrogate's embedding space that locally changes its next-token distribution. This corresponds to a first-order (tangent) direction on the surrogate's response manifold, and captures the immediate behavioral differences between the two models. By identifying and following such directions, SOMA focuses on the regions where the two models are least aligned. These mined directions then drive efficient, localized fine-tuning based on the learned soft prompts. Finally, we switch to the fine-tuned surrogate once a fast semantic closeness test is met during the service. This staged design makes local manifold approximation practical, targeted, and efficient.

## 3.1 Initialization

We initialize a soft prompt matrix $\mathbf{P} \in \mathbb{R}^{L \times d}$ on the surrogate $G$. Each row is sampled i.i.d. from a zero-mean Gaussian $\boldsymbol{p}_\ell \sim \mathcal{N}(\mathbf{0}, \sigma^2 \mathbf{I}_d)$, where $\sigma > 0$ is the initialization standard deviation. Let $\mathbf{E} \in \mathbb{R}^{d \times |\mathcal{V}|}$ be the surrogate's embedding matrix whose $v$-th column is the token embedding $\boldsymbol{e}_v \in \mathbb{R}^d$, and let $\mathrm{tok}(v)$ denote the text token for index $v \in \mathcal{V}$. Because the original model $F$ is a proprietary black-box system that does not expose its embedding layer and cannot accept continuous vectors, the soft prompt $\mathbf{P}$ cannot be fed to $F$ directly. To maintain comparable conditioning, we *verbalize* $\mathbf{P}$ into a textual prefix by nearest-neighbor projection in the surrogate's embedding space:

$$v_\ell = \arg\max_{v \in \mathcal{V}} \frac{\langle \boldsymbol{p}_\ell, \boldsymbol{e}_v \rangle}{\|\boldsymbol{p}_\ell\|_2 \, \|\boldsymbol{e}_v\|_2} \quad (\ell = 1, \dots, L_p), \quad \text{where} \quad V(\mathbf{P}) = \big(\mathrm{tok}(v_1), \dots, \mathrm{tok}(v_{L_p})\big).$$

Then let $\mathcal{D}_{t-1}$ denote the text history of the first $t-1$ turns of the original model and let $q_t$ be the query at turn $t$, the outputs of original $F$ and the surrogate $G$ at turn $t$ is given as

$$a_t^F = F\big(V(\mathbf{P}) \oplus \mathcal{D}_{t-1} \oplus q_t\big), \quad \text{and} \quad a_t^G = G\big(\mathbf{P} \oplus_{\mathrm{emb}} \mathbf{E}(\mathrm{tok}_G(\mathcal{D}_{t-1})) \oplus_{\mathrm{emb}} \mathbf{E}(\mathrm{tok}_G(q_t))\big).$$

Here, $\mathrm{tok}_G(\cdot)$ is $G$'s tokenizer, $\mathbf{E}(\cdot)$ maps tokens to embeddings, and $\oplus_{\mathrm{emb}}$ concatenates along the sequence axis in embedding space. This ensures both models are conditioned on the same $\mathcal{D}_{t-1}$ and $q_t$, with $F$ receiving the verbalized prefix $V(\mathbf{P})$ and $G$ receiving the continuous prefix $\mathbf{P}$.

## 3.2 Soft Prompt Tuning for Mining Weak Alignment Directions

In this section, we propose a differentiable loss to reliably learn soft prompts $\mathbf{P}$ that make $G$ produce outputs that differ from $F$ as much as possible on the local dialogue context. The loss consists of three parts: (i) a semantic divergence loss to ensure the token-level semantic divergence; (ii) an expectation-weight to ensure the distribution-level semantic divergence; and (iii) an anti-degeneration loss to avoid prompt mining from collapsing.

**Semantic Divergence Loss.** We first leverage *unlikelihood loss* (Welleck et al., 2019) to penalize the surrogate for assigning high probability to the exact tokens $y_F$ produced by the original model. However, this token-level formulation is limited, as it ignores close paraphrases or synonyms that carry a similar meaning. To address this, we first define the *semantic neighborhood* as below:

**Definition 3.1** (Semantic Neighborhood). *Let $\mathbf{E} \in \mathbb{R}^{d \times |\mathcal{V}|}$ be $G$'s embedding matrix with token embeddings $\boldsymbol{e}_v \in \mathbb{R}^d$. For a token $u \in \mathcal{V}$, its* semantic neighborhood *is the set of the $k$ tokens in $\mathcal{V} \setminus \{u\}$ whose embeddings have the highest cosine similarity with $\boldsymbol{e}_u$.*

In this way, the semantic neighborhood of a token $u$ can capture not only the exact token $u$ but also its closest paraphrases or synonyms in surrogate space. At turn $t$, the original model produces a text output $a_t^F$. We map this text into the surrogate's token space using $G$'s tokenizer $\boldsymbol{y}_t^F = \mathrm{tok}_G(a_t^F) = (y_{t,1}^F, \dots, y_{t,T_F(t)}^F)$. This places the original output and the surrogate distributions in the embedding space. We then define a *temperature-weighted* distribution as:

$$s_\tau\big(v \mid y_{t,i}^F\big) = \frac{\exp\big(\cos(\boldsymbol{e}_v, \boldsymbol{e}_{y_{t,i}^F})/\tau\big)}{\displaystyle\sum_{u \in \{y_{t,i}^F\} \cup \mathcal{N}_k(y_{t,i}^F)} \exp\big(\cos(\boldsymbol{e}_u, \boldsymbol{e}_{y_{t,i}^F})/\tau\big)}, \qquad k \in \mathbb{N}, \; \tau > 0.$$

This weight measures semantic proximity and allocates more mass to tokens closer to $y_{t,i}^F$. It ensures our loss penalizes soft prompts that result in not only the exact token but also its near-synonyms.

Then at turn $t$, let $S_t = \mathcal{D}_{t-1} \oplus q_t$, we feed $G$ with the token sequence $\mathrm{tok}_G(S_t)$ and $\boldsymbol{y}_t^F$, and prepend them with the soft prompt $\mathbf{P}$ at the embedding layer. This gives $G$ the same context that produced $F$'s answer. Then we run a single forward pass with $G$ on the full prefix and, with causal masking, read the logits at each position and apply softmax to obtain all next-token distributions $\{\Pi_{t,i}(\mathbf{P})\}_{i=1}^{T_F(t)}$ in one pass. These distributions are directly comparable to $F$'s tokens because both models are aligned to the same token positions defined by $G$'s tokenizer and conditioned on identical preceding text. Now we have the semantic divergence loss:

$$\mathcal{L}_{\mathrm{sem}}(\mathbf{P}; \mathcal{D}_{t-1}, q_t) = \frac{1}{T_F(t)} \sum_{i=1}^{T_F(t)} \sum_{v \in \{y_{t,i}^F\} \cup \mathcal{N}_k(y_{t,i}^F)} s_\tau\big(v \mid y_{t,i}^F\big) \Big[-\log\big(1 - \Pi_{t,i}(\mathbf{P})[v]\big)\Big], \quad (1)$$

where $\Pi_{t,i}(\mathbf{P})[v]$ is the probability that $G$ assigns to token $v$ at position $i$. Minimizing Eq 1 with respect to $\mathbf{P}$ finds soft prompts that, when concatenated with the dialogue in $G$'s embedding stream, maximize the difference between $G$ and $F$ in the local neighborhood around each original token. This produces a set of prompts that reliably surface the least-aligned small steps and thereby reveal where the two local manifolds differ most.

**Expectation-weighted Semantic Divergence Loss.** The token-level divergence in Eq. 1 can be satisfied when the surrogate $G$ reduces probability on the exact token $y_{t,i}^F$ or its top-$k$ neighbors but redistributes mass across many semantically similar tokens, so the meaning of the next-token distribution remains essentially unchanged. For example, suppose the teacher's next token is "fantastic." The surrogate gives only a small probability to "fantastic," but places most of its probability on words like "great," "excellent," and "amazing." In such cases, the meaning of the distribution is preserved even though the surface tokens differ, and a token-level penalty does not reflect this. To capture whether the entire next-token distribution still expresses the same meaning as the original, we compute the expected embedding of the surrogate's distribution and compare its direction with the embedding of the teacher's intended token. This checks whether the surrogate's overall prediction "points" toward the same meaning rather than only matching one specific token. Specifically, let $\mathbf{E} \in \mathbb{R}^{d \times |\mathcal{V}|}$ be $G$'s embedding matrix with column vectors $\boldsymbol{e}_v$ (assume $\|\boldsymbol{e}_v\|_2 = 1$), then at position $i$ of turn $t$, the expected embedding of $G$'s next-token distribution is:

$$\bar{\boldsymbol{e}}_{t,i} \;=\; \mathbf{E}^\top \Pi_{t,i}(\mathbf{P}) \;=\; \sum_{v \in \mathcal{V}} \Pi_{t,i}(\mathbf{P})[v]\, \boldsymbol{e}_v,$$

where $\Pi_{t,i}(\mathbf{P}) = \pi_G(\cdot \mid \mathcal{D}_{t-1} \oplus q_t; \mathbf{P}, a_{t,<i}^F)$ is $G$'s next-token distribution when evaluated at the same position as $F$. Then how strongly the whole distribution still aligns with the original token's meaning can be measured with $\cos\big(\bar{\boldsymbol{e}}_{t,i},\, \boldsymbol{e}_{y_{t,i}^F}\big) = \frac{\sum_v \Pi_{t,i}(\mathbf{P})[v]\, \langle \boldsymbol{e}_v, \boldsymbol{e}_{y_{t,i}^F} \rangle}{\|\bar{\boldsymbol{e}}_{t,i}\|_2}$. To strengthen the penalty exactly when the distribution-level alignment persists, we multiply the neighborhood loss by a positive, bounded weight that increases with this cosine and never reverses the loss sign:

$$w_{t,i} \;=\; 1 + \lambda \operatorname{clip}\Big( \cos\big(\bar{\boldsymbol{e}}_{t,i},\, \boldsymbol{e}_{y_{t,i}^F}\big),\, 0,\, 1\Big), \qquad \lambda \geq 0.$$

The clipping form ignores semantically opposed meaning and caps the influence of extreme alignment. The affine form preserves the base loss and only upweights when alignment is high, which keeps the objective bounded and numerically stable.

We then weight the semantic divergence loss accordingly, strengthening the penalty only when the distribution starts to drift semantically. This expectation-weighted form below therefore serves as the actual distribution-level alignment term used in the final objective:

$$\mathcal{L}_{\text{sem\_exp}}(\mathbf{P}; \mathcal{D}_{t-1}, q_t) = \frac{1}{T_F(t)} \sum_{i=1}^{T_F(t)} w_{t,i}(\mathbf{P}) \sum_{v \in \{y_{t,i}^F\} \cup \mathcal{N}_k(y_{t,i}^F)} s_\tau\big(v \mid y_{t,i}^F\big) \Big[ -\log\big(1 - \Pi_{t,i}(\mathbf{P})[v]\big)\Big].$$

$$(2)$$

Here, the neighborhood term $\sum_{v \in \{y_{t,i}^F\} \cup \mathcal{N}_k(y_{t,i}^F)} s_\tau(v \mid y_{t,i}^F) [-\log(1 - \Pi_{t,i}(\mathbf{P})[v])]$ blocks local copies, and the weight $w_{t,i}$ catches distribution-level alignment, together pushing $G$ away from $F$ in meaning rather than only in surface tokens.

**Theorem 1** (Directional recovery in the local manifold). *Let $\mathbf{J}(\mathbf{P})$ be the Jacobian of $G$'s log next-token probabilities with respect to the rows of $\mathbf{P}$ under the aligned-prefix conditioning, and let $\mathbf{C} = \mathbb{E}[\mathbf{J}^\top \mathbf{J}]$ be the empirical discrepancy Fisher matrix induced by Eq. 2 over the initial window of turns. Under local smoothness and a rank–$r$ discrepancy assumption, any minimizer of Eq. 2 produces soft prompts whose span captures at least a $(1 - \varepsilon)$ fraction of the top-$r$ eigenmass of $\mathbf{C}$, where $\varepsilon$ decreases with the neighborhood size $k$ and the number of tokens in the window.*

**Remark 1.** *The learned soft prompts implement small, structured steps on the response manifold at the current dialogue state. Their span approximates the main steps where $F$ and $G$ move differently.*

**Anti-degeneration regularizer.** Soft-prompt mining can collapse if the surrogate $G$ concentrates probability on a few high-frequency tokens, yielding repetitive or bland continuations that carry little information about where $G$ and $F$ truly differ (Li et al., 2023; Holtzman et al., 2019; Meister et al., 2023). To keep the optimization informative and consistent with local manifold exploration, we add a lightweight *training-time* diversity term that preserves entropy in $G$'s next-token distributions

near the prompt–context boundary. Using the previous $\{\Pi_{t,i}(\mathbf{P})\}_{i=1}^{T_F(t)}$, we maximize the average entropy over the last $K$ positions of the concatenated input seen by $G$:

$$H_{\text{tail}}(\mathbf{P}; t) \;=\; \frac{1}{K} \sum_{i \in \text{tail}(t,K)} \Big[ -\sum_{v \in \mathcal{V}} \Pi_{t,i}(\mathbf{P})[v] \, \log \Pi_{t,i}(\mathbf{P})[v] \Big],$$

and include the penalty $\mathcal{L}_{\text{deg}}(\mathbf{P}; t) = -\beta \, H_{\text{tail}}(\mathbf{P}; t)$ with a small $\beta > 0$. This regularizer reuses logits from the same forward pass (no extra compute), raises diversity exactly where the soft prompt interacts with the context, and prevents degenerate solutions, thereby enabling $\mathbf{P}$ to surface mean-ingful, least-aligned small steps for subsequent fine-tuning.

**Final loss and optimization.** For a minibatch of turns $\mathcal{B}$, we have the final loss as:

$$\mathcal{J}(\mathbf{P}) \;=\; \frac{1}{|\mathcal{B}|} \sum_{t \in \mathcal{B}} \Big[ \mathcal{L}_{sem\_exp}(\mathbf{P}; \mathcal{D}_{t-1}, q_t) \;-\; \beta \, H_{\text{tail}}(\mathbf{P}; t) \Big] \;+\; \lambda \, \|\mathbf{P}\|_F^2$$

Here $\beta > 0$ controls anti-degeneration strength, and $\lambda > 0$ regularizes the prompt scale. During the optimization, we minimize $\mathcal{J}(\mathbf{P})$ with AdamW, updating only $\mathbf{P}$ while freezing $G$'s weights. We also use gradient clipping to match gradient scales. To ensure efficient optimization, for each token position along the original model's answer prefix, we query an ANN index over $G$'s $L_2$-normalized token embeddings to obtain the top-$k$ neighbors of $y_{t,i}^F$. When computing the expectation-based weight, we form the expected embedding $\bar{e}_{t,i}$ using a top-$m$ truncation of $\Pi_{t,i}(\mathbf{P})$, avoiding a dense sum over $|\mathcal{V}|$. With a single forward pass over the shared prefix for all positions, SOMA's time complexity is $O\big(T_F(d \log |\mathcal{V}| + kd + md)\big)$ where $d$ the hidden size, $|\mathcal{V}|$ the vocabulary size, $T_F$ the number of prefix tokens from $F$ at which we read $G$'s next-token distribution.

### 3.3 Efficient localized fine-tuning and Inference

We use the learned soft prompts concatenated with text inputs $S_t = \mathcal{D}_{t-1} \oplus q_t$ to pair with original outputs $a_t^F$ and fine-tune the surrogate $G$ with LoRA (Hu et al., 2022) adapters. Concretely, we freeze all base weights of $G$, attach low-rank adapters to attention/MLP projections, and minimize a small objective over the local batch $\mathcal{B}_{\text{loc}}$: $\mathcal{L}_{\text{FT}}(\mathbf{\Theta}_{\text{LoRA}}) = \frac{1}{|\mathcal{B}_{\text{loc}}|} \sum_{t \in \mathcal{B}_{\text{loc}}} \text{NLL}\big(a_t^F \,\big|\, G(S_t; \mathbf{\Theta}_{\text{LoRA}})\big)$. In this paper $\phi(\cdot)$ is all-MiniLM-L6-v2 (Reimers & Gurevych, 2019).

Before switching from the original model $F$ to the fine-tuned surrogate $G$, we measure their similar-ity over the past turns using the same neighborhood-weighted cosine metric as in Eq 2. If the average similarity exceeds a threshold $\tau$, SOMA switches to $G$; otherwise, it continues querying $F$. This provides a fast semantic closeness test that the surrogate remains locally aligned with the teacher before takeover. After we switch to fine-tuned $G$, the learned soft prompts are no longer needed. To reduce the tokens needed, we further keep a fixed-budget extractive summary of the early dialogue and always append the last $K$ turns verbatim. We split past turns into sentences, embed them with the encoder $\phi(\cdot)$, maintain a running centroid of past content, and at each new turn greedily select sentences that are most relevant to the current query (high cosine to its embedding), representative of the conversation so far (high cosine to the centroid), and non-redundant with already selected sentences. We concatenate the chosen sentences to fit a token budget and use this summary with the recent turns as the prompt for $G$, which requires no calls to the original. To ensure service quality, we continuously monitor similarity between a sliding average of recent queries and the centroid to determine if a potential topic shift has occurred. That is, if this falls below a threshold or a quick recheck indicates growing divergence, we roll back for the next turn by querying $F$ once, refresh the summary on the new window, and then continue with $G$.

## 4 Theoretical Analysis

In this section, we analyze two practical knobs in SOMA to guide the selection of hyperparameters in experiments. Note that we use "local manifold" in a first-order, locally linear sense, referring to the small neighborhood around the dialogue state where the surrogate's responses vary smoothly with embedding-space perturbations, without assuming a full Riemannian or geodesic structure. Details of proof are in the Appendix D.

### 4.1 HOW MANY TURNS ARE EXPECTED TO ACCEPT THE FINE-TUNED SURROGATE?

At turn $t$, let $S_t = \mathcal{D}_{t-1} \oplus q_t$ be the shared text and let $a_t^F, a_t^G$ be the textual outputs of $F$ and the fine-tuned surrogate $G$. Let $\mathsf{Gap}(S_t) \in [0, 1]$ be a bounded discrepancy score that is 0 if the outputs are semantically identical and 1 at maximal divergence (e.g., $\mathsf{Gap}(S_t) = 1 - \cos(\phi(a_t^G), \phi(a_t^F))$ ). Define the population objective $F^\star = \mathbb{E}_{S \sim \mathcal{Q}}[\mathsf{Gap}(S)]$ and $\widehat{F}_B = \frac{1}{|B|} \sum_{S \in B} \mathsf{Gap}(S)$ for a batch $B$ of post–warm–start contexts. If the stream exhibits mild dependence, write $|B|_{\mathrm{eff}}$ for the effective size (e.g., $|B|/(1 + 2 \sum_{k \geq 1} \rho_k)$ under lag–$k$ autocorrelations $\rho_k$).

**Warm start.** We first observe $W$ turns to estimate the local context distribution $\widehat{\mathcal{Q}}_W$ (no updates), then evaluate any candidate surrogate on the warm-start window.

**Lemma 1** (Warm–start generalization). *Let $\mathrm{Gap}(S) \in [0, 1]$ and assume a weakly dependent stream with effective size $W_{\mathrm{eff}}$. Then with probability at least $1 - \delta$,*

$$\left| \frac{1}{W} \sum_{t=1}^{W} \mathrm{Gap}(S_t) - \mathbb{E}\,\mathrm{Gap}(S) \right| \leq \sqrt{\frac{2 \log(2/\delta)}{W_{\mathrm{eff}}}}.$$

.

**Post–warm–start detection.** Let a newly fine-tuned surrogate have empirical improvement $\Delta = \widehat{F}_B^{\mathrm{old}} - \widehat{F}_B^{\mathrm{new}}$ on a post–warm–start batch $B$. Assume $\Delta$ is sub–Gaussian with proxy $\sigma_\Delta^2/|B|_{\mathrm{eff}}$.

**Theorem 2** (Detection bound for switching). *Let $\Delta = \widehat{F}_B^{\mathrm{old}} - \widehat{F}_B^{\mathrm{new}}$ on a post–warm–start batch $B$, with $\Delta$ sub-Gaussian of proxy $\sigma_\Delta^2/|B|_{\mathrm{eff}}$. Fix $\varepsilon > 0$ and $\delta \in (0, 1)$. If $|B|_{\mathrm{eff}} \geq \frac{2\sigma_\Delta^2}{\varepsilon^2} \log \frac{1}{\delta} + \frac{2}{3\varepsilon} \log \frac{1}{\delta}$, then $\Pr(\Delta \geq \varepsilon) \geq 1 - \delta$.*

**Corollary 1** (Switching rule). *If $\widehat{F}_B^{\mathrm{old}} - \widehat{F}_B^{\mathrm{new}} \geq \varepsilon$ for a batch meeting Theorem 2, switching is justified at level $1 - \delta$.*

**Corollary 2** (Decision error). *Combining Lemma 1 and Theorem 2, the total decision error from warm–start approximation and batch detection is at most $\eta + \varepsilon$ with confidence $1 - 2\delta$.*

### 4.2 HOW MANY SOFT–PROMPT CANDIDATES PER ITERATION?

Assume the objective restricted to the local tangent near a reference prompt $\mathbf{P}_0$ admits a quadratic model with positive semidefinite curvature $\mathbf{H}_T$ whose dominant energy lies in an $r_{\mathrm{act}}$–dimensional active subspace ($r_{\mathrm{act}} \ll d$). Let $u_1$ be the leading eigenvector of $\mathbf{H}_T$ in this subspace. Each soft–prompt candidate is a unit vector $u_m$ sampled uniformly from this active subspace.

**Theorem 3** (Coverage of the best local direction). *Let the active subspace have dimension $r_{\mathrm{act}} \geq 1$, and let $\mathbf{u}_1$ be the target unit direction. Draw $M$ i.i.d. unit candidates $\{\mathbf{u}_m^{(c)}\}_{m=1}^M$ uniformly on $\mathbb{S}^{r_{\mathrm{act}}-1}$. For any angle threshold $\theta \in (0, \pi/2]$,*

$$\Pr\Big( \min_{1 \leq m \leq M} \angle(\mathbf{u}_m^{(c)}, \mathbf{u}_1) \leq \theta \Big) \geq 1 - \Big( 1 - (\sin \theta)^{r_{\mathrm{act}}-1} \Big)^M.$$

**Lemma 2** (Directional suboptimality). *Let $\mathbf{H}_T \succeq \mathbf{0}$ be the curvature on the active subspace with eigenvalues $\lambda_1 \geq \cdots \geq \lambda_{r_{\mathrm{act}}} \geq 0$ and top eigenvector $\mathbf{u}_1$. For any unit $\widehat{\mathbf{u}}$ with $\angle(\widehat{\mathbf{u}}, \mathbf{u}_1) \leq \theta$,*

$$\widehat{\mathbf{u}}^\top \mathbf{H}_T \widehat{\mathbf{u}} \geq \lambda_1 \cos^2 \theta, \qquad \text{hence} \qquad \lambda_1 - \widehat{\mathbf{u}}^\top \mathbf{H}_T \widehat{\mathbf{u}} \leq \lambda_1 \sin^2 \theta.$$

**Corollary 3.** *To ensure coverage probability at least $1 - \delta$ at angle $\theta$, $M \geq \frac{\log(1/\delta)}{\log\big(1 - (\sin \theta)^{r_{\mathrm{act}}-1}\big)}$.*

## 5 EMPIRICAL STUDIES OF THE EFFECTIVNESS OF SOMA

In this section, we empirically evaluate the effectiveness of SOMA by addressing the following questions: **RQ1**: How well does SOMA perform against baselines? **RQ2**: How much efficiency does SOMA improve? **RQ3**: How does each component of SOMA affect the performance?

**Datasets.** We evaluate on six multi-turn datasets: ShareGPT (Chen et al., 2024a), ReMeDi (Yan et al., 2022), Craigslist (He et al., 2018), Multi-Char (agentlans, 2024), MATH (Hendrycks et al., 2021), and MT-Bench (Zheng et al., 2023). More details are shown in Appendix B.1

Table 1: Similarity percentage to the original model across six datasets for LLaMA family.

|  | ShareGPT | ReMeDi | Craigslist | Multi-Char | MATH | MT-Bench | Avg |
|---|---|---|---|---|---|---|---|
| Surrogate | $79.2 \pm 2.18$ | $82.7 \pm 1.95$ | $74.3 \pm 2.36$ | $70.8 \pm 1.73$ | $66.2 \pm 2.91$ | $77.5 \pm 2.04$ | $75.1 \pm 5.98$ |
| History-Prefix | $86.1 \pm 1.67$ | $87.9 \pm 1.55$ | $82.4 \pm 1.72$ | $84.7 \pm 1.69$ | $80.3 \pm 3.83$ | $87.2 \pm 1.58$ | $84.8 \pm 2.94$ |
| History-FT | $93.4 \pm 2.12$ | $91.8 \pm 2.09$ | $90.3 \pm 1.23$ | $89.1 \pm 2.23$ | $87.6 \pm 2.26$ | $92.4 \pm 1.14$ | $90.8 \pm 2.18$ |
| LLMLingua-2 | $84.6 \pm 1.72$ | $86.4 \pm 1.63$ | $80.8 \pm 1.91$ | $82.9 \pm 1.58$ | $78.1 \pm 2.77$ | $85.3 \pm 1.66$ | $83.0 \pm 3.03$ |
| RouteLLM | $95.3 \pm 1.44$ | $92.5 \pm 1.07$ | $91.0 \pm 1.86$ | $91.4 \pm 1.23$ | $89.6 \pm 1.95$ | $93.2 \pm 1.12$ | $92.2 \pm 1.78$ |
| **SOMA** | $\mathbf{96.4 \pm 1.91}$ | $\mathbf{93.2 \pm 0.98}$ | $\mathbf{91.9 \pm 2.49}$ | $\mathbf{92.3 \pm 1.05}$ | $\mathbf{90.7 \pm 1.12}$ | $\mathbf{94.1 \pm 0.91}$ | $\mathbf{93.1 \pm 1.99}$ |

**Models and baselines.** For LLaMA (Touvron et al., 2023) we use LLaMA-3.1-70B (original) and LLaMA-2-7B (surrogate); for Qwen (Bai et al., 2023) we use Qwen-3-8B (original) and Qwen-3-0.6B (surrogate). Baselines: (i) Original; (ii) Surrogate; (iii) History-Prefix (surrogate with the original's full history up to the current turn); (iv) History-FT (fine-tune surrogate on the original's history and serve latter turns); (v) single-model approach: LLMLingua-2 (compressing chat histories (Pan et al., 2024)); (vi) Multi-model approach: RouteLLM routing (original for complex and surrogate for simple) (Ong et al., 2024). Additional details are provided in Appendix B.3.

**Evaluation Metrics.** Response quality is evaluated by the similarity of each method's output to the original model's output. We use three LLM judges—GPT-OSS (OpenAI, 2025), DeepSeek-V3, and Gemma-2-27B as in Appendix B.5, and report the average rating across judges to reduce single-judge bias. Efficiency is measured by average tokens per dialogue and throughput (tokens/s).

**Implementation.** We instantiate all knobs directly based on Section 4. First, the switching window $W$ and the acceptance batch size follow the detection bound (Thm. 2) together with warm-start generalization (Lemma 1). Concretely, after a warm start of $W$ turns large enough to make the generalization error $O(W_{\text{eff}}^{-1/2})$ small, we choose $|B|_{\text{eff}}$ so that the bound on $\Pr(\Delta \geq \varepsilon)$ exceeds 0.95 for the empirically estimated $\sigma_\Delta$. The number of parallel soft-prompt candidates $M$ is set via the spherical-cap coverage guarantee (Thm. 3) and the directional suboptimality bound (Lemma 2). The cosine gate is calibrated to the target error budget using Cor. 2: we select a threshold that limits false switches to $\leq 5\%$ on the warm-start window and require $m = 2$–3 consecutive hits for stability. Further details appear in Appendix B.2.

## 5.1 EXPERIMENTAL RESULTS

**SOMA consistently performs similarly to the original model and outperforms baselines.** As shown in Table 1, SOMA has the highest similarity to the original's responses consistently. Compared with SOMA, the Surrogate alone has limited ability and struggles in complex turns; History-Prefix supplies the full history of original model but the surrogate itself still has limited ability; History-FT trains on full original history but can only learn superficial phrasing since the supervision information is limited; LLMLingua-2 sometimes neglects important context details; RouteLLM switches models but never improves the small model itself. Dataset-wise, SOMA's gains are largest on MATH and Multi-Character compared with surrogate, where later turns require careful carry-over of constraints and multi-step reasoning, and our designed expectation-weighted divergence and the anti-degeneration guard make the mined cases informative here. Improvements are smaller on MT-Bench and ShareGPT, where some baselines perform well due to easier queries, but SOMA can consistently win, as SOMA improved the surrogate. More results are in Appendix E.1.

**SOMA is more efficient than baselines.** As shown in Figure 2, SOMA consistently outperforms baselines in terms of efficiency. The reason is that SOMA stops resending the growing history and serves the remaining turns with the adapted small model with a compressed context. Original and RouteLLM are the most expensive as both repeatedly transmit long contexts to a large model, and RouteLLM also pays routing overhead when it escalates. History-Prefix remains high since it forwards the full history to the surrogate every turn; History-FT saves some tokens but still carries long prompts. LLMLingua-2 compresses history and helps, yet summaries + control

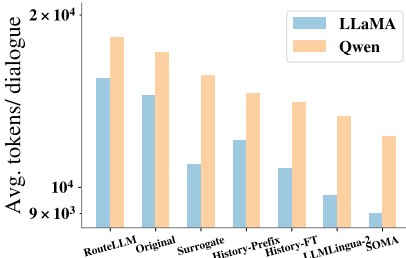

Figure 2: Average tokens per dialogue shows SOMA uses the fewest tokens and reduces compute/API cost.

prompts keep usage above SOMA. Surrogate only reduces tokens versus Original but lacks the switch-and-adapt step, so it cannot drop the history tail as aggressively. Overall, SOMA achieves the best cost efficiency on both LLaMA and Qwen. The results on throughput are presented in Appendix E.2.

**Ablation Study.** As shown in Figure 3, the full SOMA achieves the highest similarity. Removing the anti-degeneration loss (w/o ADL) consistently drops performance, while removing both the expectation-weight and ADL (w/o ExpW+ADL) yields the larger decline, showing that entropy regularization prevents collapse and distribution-level weighting is critical. The ablation study on Qwen is presented in Appendix E.3

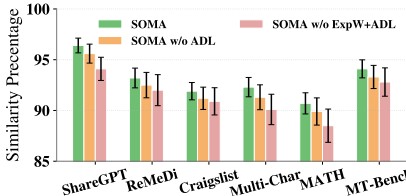

Figure 3: Ablation Study (LLaMA).

## 6 RELATED WORK

**Efficient LLM for Multi-turn Dialogues.** Recent efforts to improve multi-turn LLM serving efficiency fall into two main paradigms. The first involves **single-model approaches** that reduce context length or reuse computation. These include *summarization and context compression* (Wang et al., 2025; Chen et al., 2024b; Xiao et al., 2024), *memory augmentation* (Melz, 2023; Gutiérrez et al., 2024), and *caching and attention reuse* (Gao et al., 2024; Jeong & Ahn, 2025; Anthropic, 2024). While effective in lowering per-turn cost, these methods still rely on repeated large-model inference, which also leads to high API expenses, latency, and GPU demand. Moreover, they may truncate or underutilize dialogue context, harming performance on complex tasks. The second paradigm adopts **multi-model approach**, using smaller models for simple queries and escalating difficult ones to larger LLMs (Behera et al., 2025; Schick et al., 2023; Ding et al., 2024), typically via model routing (Shnitzer et al., 2023) and distillation (Hinton et al., 2015). However, pre-trained small models often generalize poorly on complex multi-turn dialogues, while switching models adds inefficiency.

**Local Manifold Approximation.** Local manifold approximation techniques aim to exploit the manifold hypothesis by modeling high-dimensional data as lying on locally low-dimensional subspaces, enabling more efficient representation and inference. Classical approaches such as Locally Linear Embedding (LLE) (Roweis & Saul, 2000) and Local Tangent Space Alignment (LTSA) (Zhang et al., 2007) approximate local neighborhoods through linear projections, while kernel-based methods like Laplacian eigenmaps (Belkin & Niyogi, 2003) and diffusion maps (Coifman & Lafon, 2006) preserve local geometric structure via nonlinear embeddings. Recent work has extended these ideas using deep learning. For instance, neural network–based tangent space estimators (Sun et al., 2020) and local contrastive learning methods (Xiong et al., 2020; Zeng et al., 2021) enable the extraction of manifold-aware representations in complex domains. In computer vision, local manifold modeling underpins point cloud upsampling (Fang & Wang, 2025) by fitting Gaussian patches to local regions, while in representation learning, neighbor-preserving mappings such as t-SNE (Van der Maaten & Hinton, 2008) and UMAP (McInnes et al., 2018) uncover latent structure by maintaining local proximity. In graph-based learning, manifold-regularized GNNs (Ngo & Vo, 2023) exploit smoothness over graph-induced manifolds to enhance generalization. Despite their effectiveness across domains, local manifold approximation remains largely unexplored in the context of efficient multi-turn LLM serving, where dynamically adapting smaller models to the local reasoning manifold conditioned on dialogue history presents a promising and under-investigated direction.

## 7 CONCLUSION

In this paper, we first identified a long-tail pattern in multi-turn dialogues: early turns contain most of the goal, constraint, and contextual information, while later turns rely heavily on the accumulated state rather than introducing new structure. Building on this observation, we introduced SOMA, a framework that leverages these early-turn interactions to mine informative soft prompts, apply lightweight localized fine-tuning to adjust the surrogate model in the immediate context, and use a simple semantic-drift gate to determine when it is safe to switch to the smaller model. Extensive experiments show that SOMA consistently improves surrogate–teacher similarity and substantially reduces the number of calls to the large model, and these empirical behaviors match the predictions of our theoretical analysis. Taken together, our paper offers a practical and effective path toward efficient multi-turn LLM serving and points to promising opportunities for future work.

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

## A    NOTATIONS

This section summarizes all notations used throughout this paper.

Table 2: Notation summary.

| Symbol | Meaning |
|---|---|
| $\mathcal{D}_k$ | Dialogue prefix $\{(q_t, a_t)\}_{t=1}^{k}$ |
| $q_{\leq t}$ | Full history up to turn $t$ |
| $F$, $G$ | Original (large) model; surrogate (small) model |
| $\mathbf{h}_t = f_M(q_{\leq t})$ | Hidden state at turn $t$ for model $M$ |
| $\mathcal{H}_k$ | $\{\mathbf{h}_1, \ldots, \mathbf{h}_k\}$, first-$k$ hidden states |
| $\mathcal{M}_k^M$ | Local response manifold induced by $\mathcal{D}_k$ under $M$ |
| $\mathcal{V}$, $E$ | $G$'s vocabulary; embedding matrix $E \in \mathbb{R}^{d \times |\mathcal{V}|}$ |
| $\Pi_t$ | Next-token distribution of $G$ at turn $t$; entry $\Pi_t[v]$ |
| $\mathbf{e}_v$ | Embedding of token $v$ (column of $E$) |
| $\mathbf{P} \in \mathbb{R}^{L \times d}$ | Soft prompt (row-wise prompt tokens in embedding space) |
| $\mathsf{V}(\mathbf{P})$ | Verbalization for prompt $\mathbf{P}$ |
| $N_k(u)$ | $k$ nearest neighbors of token $u$ by cosine in $E$ |
| $\bar{\mathbf{e}}_{t,i}$ | Expected embedding $\bar{\mathbf{e}}_{t,i} = E^\top \Pi_{t,i}$ at position $i$ |
| $w_{t,i}$ | Expectation-weighted factor for semantic divergence loss |
| $|B|_{\text{eff}}$ | Effective batch size (accounts for dependence) |
| $\Delta$ | Empirical improvement (old–new) on a post-warm-start batch |

## B    REPRODUCIBILITY

In this section, we introduce the details of the experiments in this paper for reproducibility. At the same time, we have uploaded all necessary code to our GitHub repository to reproduce the results presented in this paper: `https://anonymous.4open.science/r/SOMA-D377`. All major experiments are encapsulated as shell scripts, which can be conveniently executed. We introduce the details for reproducibility in the subsections below.

### B.1    REAL-WORLD DATASETS

In this section, we briefly present the real-world graph datasets used in this paper, and all these datasets are commonly used datasets in multi-turn conversation tasks. *ShareGPT* (Chen et al., 2024a) is a large-scale collection of high-quality image–text conversations and captions. *ReMeDi* (Yan et al., 2022) is a multi-domain Chinese medical dialogue corpus of doctor–patient conversations. *Craigslist* (He et al., 2018) contains multi-turn buyer–seller negotiation chats from Craigslist, enabling study of bargaining strategies and goal-directed dialogue. *Multi-Char* (agent-lans, 2024) provides multi-character conversational scenarios with role specifications to evaluate coordination and role consistency in multi-party dialogue. *MATH* (Hendrycks et al., 2021) is a benchmark of competition-style math problems with step-by-step solutions designed to assess mathematical reasoning in language models. *MT-Bench* (Zheng et al., 2023) is a multi-turn benchmark to assess response quality across diverse tasks. In this study, we filter out the non-context-dependent dialogues in these datasets.

### B.2    IMPLEMENTATION OF SOMA

We implement SOMA based on PyTorch with HuggingFace Transformers, serve inference via vLLM with FlashAttention (Shah et al., 2024), and run on one node with $4\times$ 80G A100 GPUs. Soft–prompt tuning optimizes only the prompt tensor $\mathbf{P} \in \mathbb{R}^{L \times d}$ on the surrogate $G$ using AdamW, cosine decay, gradient clipping, and a KV cache for a single forward pass per turn. The objective combines unlikelihood on a semantic neighborhood, an expectation–weighted penalty using the truncated top–$m$ expectation, and a light anti-degeneration entropy term. The mined prompt–response pairs are then used to adapt $G$ with LoRA on attention projections (rank $r$), keeping the base weights frozen; early stopping is triggered by validation divergence. At inference, a

cosine closeness gate with a lightweight sentence encoder decides a one–time switch to the adapted surrogate; the window $W$ and acceptance batch $|B|$ follow the detection bound, the number of parallel candidates $M$ follows the coverage guarantee, and $(k, m)$ follow the efficiency analysis. Prompt length $L \in \{4, 8, 16, 32, 64\}$; learning rate $\eta \in [1 \times 10^{-4}, 5 \times 10^{-3}]$; AdamW weight decay $\lambda_{\text{wd}} \in [0, 10^{-2}]$; gradient clip $\in [0.5, 1.0]$; neighborhood size $k \in \{20, 50, 100\}$; expectation truncation $m \in \{50, 100, 200\}$; temperature $\tau \in [0.4, 1.2]$; expectation weight $\lambda \in [0.5, 2.0]$; anti–degeneration weight $\beta \in [0.02, 0.15]$; LoRA rank $r \in \{4, 8, 16, 32\}$ with scale $\alpha_{\text{lora}} \in \{4, 8, 16, 32\}$; LoRA LR $\eta_{\text{lora}} \in [5 \times 10^{-5}, 2 \times 10^{-3}]$; warm–start window $W \in [3, 12]$ turns; acceptance batch $|B| \in [6, 24]$ contexts; parallel candidates $M \in \{3, 4, 5\}$; switch threshold $\varepsilon \in [0.05, 0.12]$ with $m_{\text{cons}} \in \{2, 3\}$.

### B.3 Implementation of Baselines

**Original**: We query the large model (family-appropriate: LLaMA-3.1-70B or Qwen-3-8B) with the full dialogue history at every turn to get the responses.

**Surrogate**: We query the small model (LLaMA-2-7B or Qwen-3-0.6B) with the full dialogue history at every turn to get the responses.

**History-Prefix**: The surrogate receives the entire conversation produced by the Original up to $t-1$ and generates $a_t$; no parameter updates are performed. The switching is the same as SOMA.

**History-FT**: We fine-tune the surrogate on $(S_t, a_t^F)$ pairs where $S_t$ is the Original's full context up to $t$ and $a_t^F$ is the next reply, using LoRA on attention projections with early stopping; inference then runs the fine-tuned surrogate without the Original. The switching is the same as SOMA.

**LLMLingua-2** (Pan et al., 2024): We compress the Original's history at each turn with LLMLingua-2 and feed the compressed summary to the surrogate; we follow the authors' recommended settings.

**RouteLLM** (Ong et al., 2024): We adopt the released router to choose between the Original and the surrogate per turn (complex vs. simple queries), following the original settings from the paper.

### B.4 Data Filtering

To ensure that SOMA is evaluated on context-dependent dialogues, we prompt multiple strong LLMs, including GPT-OSS, DeepSeek-V3, and Gemma-2-27B, with the classification prompt shown in Figure 4, and retain only conversations that all three models recognize as context-dependent. This avoids model-specific biases and yields a high-quality subset of dialogues where later turns meaningfully depend on earlier turns, matching the use cases in this paper.

### B.5 Implementation of LLM Judge

The prompt for the LLM judge to evaluate the response similarity score is shown in Figure 5 following the recommendation from (Bai et al., 2024).

### B.6 Packages Required for Implementation

We perform all experiments on a server equipped with Nvidia A6000 GPUs. Below we list the key packages and their versions used in our implementation:

- **Python** == 3.10
- **pytorch** == 2.8.0 + CUDA 12.8
- **torchvision** == 0.19.0
- **torchaudio** == 2.8.0
- **numpy** == 1.26.x
- **pandas** == 2.2.x
- **scipy** == 1.12.x
- **cmake** == 3.28+
- **ninja** == 1.11+

You are an expert conversation analyst.

Task – Classification:
* **context-dependent**: The response of every turn relies on information from earlier turns (the conversation is a single chain of related reasoning steps).
* **context-independent**: The response of every turn is independent of earlier turns (the conversation is a collection of independent questions and answers).

Conversation (verbatim):
{dialogue}

Question: Is the conversation context-dependent or context-independent?
Please answer with exactly one word:
Either **context-dependent** or **context-independent** — nothing else.

Figure 4: Instructions for the LLM to filter context-dependent dialogue.

- **ipython** == 8.x
- **psutil** == 5.9+
- **vllm** == 0.10.2
- **transformers** == 4.56.1
- **accelerate** == 1.9.0
- **bitsandbytes** == 0.46.1
- **sentencepiece** == 0.2.0
- **tiktoken** == 0.11.0
- **einops** == 0.8.1
- **datasets** == 4.0.0
- **huggingface-hub** == 0.34.2
- **safetensors** == 0.5.3
- **ray** == 2.49.1
- **scikit-learn** == 1.7.1
- **fastapi** == 0.116.2
- **uvicorn** == 0.35.0

## C  MATHEMATICAL CONCEPTS

Here we provide a more comprehensive view of the relevant concepts that can be helpful to understand the idea of stratification as discussed in the main body of the paper.

**Definition C.1** (Preimage). *Let $f : X \mapsto Y$ be a function from a set $X$ (domain) to a set $Y$ (codomain). For any subset $N \subseteq Y$, the* preimage *of $N$ under $f$, denoted $f^{-1}(N)$, is defined as:*

$$f^{-1}(N) = \{x \in X \mid f(x) \in N\}.$$

918
919
920
921
922
923
924
925
926
927
928
929
930
931
932
933
934
935
936
937
938
939
940
941
942
943
944
945

> You are an impartial evaluator.
> You are evaluating the AI assistant's capabilities in solving a problem posed by 'Human'.
> Assess the assistant's performance based on the following criteria:
>
> 1. Accuracy: Verify the correctness of the AI assistant's answer against the ground truth
> (reference solution).
> 2. Reasoning: Assess the completeness, clarity, and logical soundness of the step-by-step
> reasoning process.
> 3. Context Integration: Consider whether the assistant integrates relevant prior dialogue or
> context that influences the solution.
> 4. Communication: Appraise how clearly and effectively the assistant communicates its
> reasoning to aid understanding.
>
> Score Guidelines (0.0 to 1.0):
> - 1.0: Completely correct answer and meticulously clear, step-by-step reasoning that is
> logically sound and instructional.
> - 0.7–0.9: Correct answer with a well-articulated reasoning process that includes the necessary
> steps and promotes understanding.
> - 0.4–0.6: Partially correct answer with some minor reasoning flaws or omissions.
> - 0.0–0.3: Incorrect answer and/or poor reasoning that lacks clarity or logic.
>
> Only respond with a single score between 0.0 and 1.0. Do not include any explanation.
> ---
>
> Reference Solution:
> {Original Response}
> Candidate Answer:
> {Surrogate Response}
>
> Score:

Figure 5: Instructions for the LLM judge to evaluate the response similarity.

In other words, $f^{-1}(N)$ consists of all elements in the domain $X$ that are mapped into the subset $N$ of the codomain $Y$.

**Definition C.2** (Metric Space). *A set $X$, whose elements are called points, is said to be a* metric space *if for any two points $p, q \in X$, there is an associated real number $d(p, q)$, called the* distance *from $p$ to $q$, such that:*

1. *$d(p, q) \geq 0$, and $d(p, q) = 0 \iff p = q$;*

2. *$d(p, q) = d(q, p)$ (symmetry);*

3. *$d(p, q) \leq d(p, r) + d(r, q)$ for any $r \in X$ (triangle inequality).*

*Any function satisfying these properties is called a* distance function*, or a* metric.

**Definition C.3** (Neighborhood). *Let $X$ be a metric space. A set $N_r(p) \subset X$ is called a* neighborhood *of a point $p \in X$ if it consists of all points $q \in X$ such that $d(p, q) < r$ for some radius $r > 0$. The number $r$ is called the* radius *of the neighborhood.*

**Definition C.4** (Continuous). *Let $X$ and $Y$ be two topological spaces. A function $f : X \mapsto Y$ is* continuous *if for each point $x \in X$ and each neighborhood $N$ of $f(x)$ in $Y$, the set $f^{-1}(N)$ is a neighborhood of $x \in X$.*

**Definition C.5** (Topological Equivalence or Homeomorphism). *A function $h : X \mapsto Y$ is called a* homeomorphism *if it is one-to-one, continuous, and has a continuous inverse function. When such a function exists, $X$ and $Y$ are called* homeomorphic *(or* topologically equivalent*) spaces.*

**Definition C.6** (Open Set). *A subset $U \subseteq X$ is called an* open set *if for every point $p \in U$, there exists a neighborhood $N_r(p) \subseteq U$. That is, each point in $U$ has some "wiggle room" around it that still lies entirely within $U$.*

**Definition C.7** (Countable Base (Second Countability)). *Let $X$ be a topological space. A collection $\mathcal{B}$ of open subsets of $X$ is called a* base *(or* basis*) for the topology on $X$ if for every open set $U \subseteq X$ and every point $x \in U$, there exists a set $B \in \mathcal{B}$ such that*

$$x \in B \subseteq U.$$

*If there exists a base $\mathcal{B}$ that is* countable*, then the space $X$ is said to be* second countable *or to have a* countable base.

**Definition C.8** (Hausdorff Space)**.** *A topological space with the property that two distinct points can always be surrounded by disjoint open sets is called a* Hausdorff space.

Essentially, Hausdorff spaces are the spaces where any two points being "far off" is defined.

**Definition C.9** (Manifold)**.** *A manifold of dimension $n$ is a second-countable Hausdorff topological space in which each point has a neighborhood homeomorphic to Euclidean space $\mathbb{R}^n$.*

**Definition C.10** (Smooth Manifold)**.** *A smooth manifold is a manifold $\mathcal{M}$ equipped with a collection of coordinate charts (i.e., homeomorphisms $\varphi : U \to \mathbb{R}^n$ for open sets $U \subseteq \mathcal{M}$) such that all* transition maps *between overlapping charts,*

$$\varphi_j \circ \varphi_i^{-1} : \varphi_i(U_i \cap U_j) \to \varphi_j(U_i \cap U_j),$$

*are infinitely differentiable (i.e., $C^\infty$). This structure is known as a* smooth atlas*, and it allows calculus to be performed on the manifold.*

# D  THEORETICAL RESULTS AND PROOFS

We first model weak dependence in the dialogue stream via an *effective sample size*. For a bounded, stationary sequence $\{Z_t\}$ with lag-$\ell$ autocorrelation $\rho(\ell)$, define

$$N_{\text{eff}} := \frac{N}{1 + 2\sum_{\ell=1}^{N-1}\left(1 - \frac{\ell}{N}\right)\rho(\ell)} \quad \in (0, N].$$

when $\rho(\ell) \equiv 0$, $N_{\text{eff}} = N$; positive correlation reduces $N_{\text{eff}}$.

**Directional Recovery for the Expectation–Weighted Loss**

**Statement (Theorem 1).** Under local smoothness, bounded expectation weights in Eq. equation 2, and a rank–$r$ discrepancy for the discrepancy Fisher $\mathbf{C} = \mathbb{E}[\mathbf{J}(\mathbf{P})^\top \mathbf{J}(\mathbf{P})]$, any minimizer $\widehat{\mathbf{P}}$ of Eq. equation 2 has row span that captures at least a $(1 - \varepsilon)$ fraction of the top-$r$ eigenmass of $\mathbf{C}$, with $\varepsilon \to 0$ as the window size grows and neighborhood size $k$ increases within the local region.

**Proof of Theorem 1.  Step 1 (Per-event loss and local expansion).** Index events by $z = (t, i)$ (turn and position). Let the per-event loss be

$$\ell(\mathbf{P}; z) = w(z) \sum_{v \in \mathcal{S}(z)} s_\tau(v \mid z) \left[ -\log\big(1 - \pi_G(v \mid z; \mathbf{P})\big) \right], \quad \mathcal{S}(z) = \{y_{t,i}^F\} \cup \mathcal{N}_k(y_{t,i}^F).$$

Write $p_v(\mathbf{P}; z) = \pi_G(v \mid z; \mathbf{P})$ and expand around $\mathbf{P} = \mathbf{0}$. Using the chain rule for the softmax parameterization,

$$\log(1 - p_v) = \log(1 - p_v|_{\mathbf{0}}) - \frac{1}{1 - p_v|_{\mathbf{0}}}\left(\nabla p_v\right)^\top \text{vec}(\mathbf{P}) - \frac{1}{2}\frac{p_v|_{\mathbf{0}}}{(1 - p_v|_{\mathbf{0}})^2}\left(\text{vec}(\mathbf{P})^\top \nabla \log p_v\right)^2 + o(\|\mathbf{P}\|^2),$$

where $\nabla$ denotes gradient w.r.t. $\text{vec}(\mathbf{P})$ and we used the softmax identity $\nabla p_v = p_v \nabla \log p_v$. Summing over $v \in \mathcal{S}(z)$ with bounded weights $w(z)s_\tau(\cdot \mid z)$ cancels the *linear* term due to local stationarity under aligned-prefix conditioning (the gradient at $\mathbf{0}$ integrates to zero across the semantic neighborhood; this is the standard property behind Gauss–Newton/Fisher approximations). Thus the second-order term dominates:

$$\ell(\mathbf{P}; z) = \frac{1}{2}\left\|\mathbf{J}_z \text{vec}(\mathbf{P})\right\|_2^2 + o(\|\mathbf{P}\|^2),$$

where $\mathbf{J}_z$ stacks rows $\sqrt{w(z)s_\tau(v \mid z)}\,\nabla \log p_v(\mathbf{0}; z)^\top$ for $v \in \mathcal{S}(z)$.

**Step 2 (Summation over the window and Fisher form).** Sum over $z$ in the initial window and take expectation (over the empirical distribution of aligned-prefix contexts). We obtain the Gauss–Newton surrogate

$$\mathcal{L}(\mathbf{P}) = \frac{1}{2}\text{vec}(\mathbf{P})^\top \mathbf{C}\,\text{vec}(\mathbf{P}) + o(\|\mathbf{P}\|^2), \qquad \mathbf{C} = \mathbb{E}\big[\mathbf{J}(\mathbf{0})^\top \mathbf{J}(\mathbf{0})\big],$$

which is a discrepancy Fisher matrix with importance weights folded into $\mathbf{J}$.

**Step 3 (Row-span parametrization and Ky Fan).** Let $\mathbf{P} \in \mathbb{R}^{L \times d}$. Any $\mathbf{P}$ factorizes as $\mathbf{P} = \mathbf{A}\mathbf{U}^\top$ with $\mathbf{U} \in \mathbb{R}^{d \times L}$ having orthonormal columns spanning the row space and $\mathbf{A} \in \mathbb{R}^{L \times L}$. Then

$$\mathrm{vec}(\mathbf{P}) = (\mathbf{U} \otimes \mathbf{I}_L)\,\mathrm{vec}(\mathbf{A}), \quad \mathcal{L}(\mathbf{P}) = \frac{1}{2}\,\mathrm{vec}(\mathbf{A})^\top \left(\mathbf{U}^\top \mathbf{C}\mathbf{U} \otimes \mathbf{I}_L\right)\mathrm{vec}(\mathbf{A}) + o(\|\mathbf{A}\|^2).$$

Including the ridge $\lambda\|\mathbf{P}\|_F^2 = \lambda\|\mathbf{A}\|_F^2$ from the main loss, the minimum over $\mathbf{A}$ for fixed $\mathbf{U}$ is proportional to $\mathrm{Tr}(\mathbf{U}^\top \mathbf{C}\mathbf{U})$. Maximizing $\mathrm{Tr}(\mathbf{U}^\top \mathbf{C}\mathbf{U})$ over $\mathbf{U}^\top \mathbf{U} = \mathbf{I}_L$ is solved by the top-$L$ eigenvectors of $\mathbf{C}$ (Ky Fan's variational principle). Hence the *optimal row span* equals the top-$L$ eigenspace of $\mathbf{C}$.

**Step 4 (Empirical approximation and eigenspace stability).** We work with an empirical Fisher $\widehat{\mathbf{C}}$ formed from finitely many events and finite $k$. Under bounded weights and local smoothness, $\|\widehat{\mathbf{C}} - \mathbf{C}\|_{\mathrm{op}} \to 0$ as the window size grows; increasing $k$ within the local isotropy region reduces variance and retains locality. Davis–Kahan perturbation then gives that the top-$r$ eigenspaces of $\widehat{\mathbf{C}}$ and $\mathbf{C}$ are close, with principal angles bounded by $O(\|\widehat{\mathbf{C}} - \mathbf{C}\|_{\mathrm{op}}/\mathrm{gap})$, where $\mathrm{gap}$ is the spectral gap below $\lambda_r(\mathbf{C})$. Consequently, the row span of any empirical minimizer $\widehat{\mathbf{P}}$ captures at least a $(1 - \varepsilon)$ fraction of the top-$r$ eigenmass of $\mathbf{C}$ with $\varepsilon \to 0$ as the window grows and $k$ increases up to the local isotropy scale. $\qquad\square$

**When to Switch: Warm–Start Generalization and Batch Detection.**

**Statement (Lemma 1).** With probability at least $1 - \delta$,

$$\left|\widehat{F}_W - F^\star\right| \leq \sqrt{\frac{2\log(2/\delta)}{W_{\mathrm{eff}}}}, \qquad \widehat{F}_W = \frac{1}{W}\sum_{t=1}^{W}\mathsf{Gap}(S_t), \quad F^\star = \mathbb{E}_{S \sim \mathcal{Q}}[\mathsf{Gap}(S)].$$

**Proof of Lemma 1.** **Step 1 (Centering and boundedness).** Let $Z_t = \mathsf{Gap}(S_t) - F^\star \in [-1, 1]$ with $\mathbb{E}[Z_t] = 0$. Then $\widehat{F}_W - F^\star = \frac{1}{W}\sum_{t=1}^{W} Z_t$.

**Step 2 (Variance proxy under dependence).** For stationary $\{Z_t\}$ with autocovariance $\gamma(\ell) = \mathrm{Cov}(Z_t, Z_{t+\ell})$, we have

$$\mathrm{Var}\left(\frac{1}{W}\sum_{t=1}^{W} Z_t\right) = \frac{1}{W^2}\sum_{t,s=1}^{W}\gamma(|t-s|) = \frac{1}{W}\left(\gamma(0) + 2\sum_{\ell=1}^{W-1}\left(1 - \frac{\ell}{W}\right)\gamma(\ell)\right).$$

Let $\sigma^2 := \gamma(0) \leq 1/4$ (since $Z_t \in [-1, 1]$) and $\rho(\ell) := \gamma(\ell)/\gamma(0)$ when $\gamma(0) > 0$. Then

$$\mathrm{Var}\left(\frac{1}{W}\sum_{t=1}^{W} Z_t\right) \leq \frac{\sigma^2}{W}\left(1 + 2\sum_{\ell=1}^{W-1}\left(1 - \frac{\ell}{W}\right)\rho(\ell)\right).$$

Define the effective size $W_{\mathrm{eff}}$ as in the preliminaries. Then $\mathrm{Var}\left(\frac{1}{W}\sum_{t=1}^{W} Z_t\right) \leq \sigma^2/W_{\mathrm{eff}}$.

**Step 3 (Concentration with effective size).** A Hoeffding/Rio inequality for bounded, weakly dependent sequences yields

$$\Pr\left(\left|\widehat{F}_W - F^\star\right| \geq u\right) \leq 2\exp\left(-\frac{2u^2 W_{\mathrm{eff}}}{(b-a)^2}\right), \quad a = -1,\ b = 1.$$

Thus $\Pr(|\widehat{F}_W - F^\star| \geq u) \leq 2\exp(-2u^2 W_{\mathrm{eff}})$. Set the right-hand side to $\delta$ and solve for $u$ to obtain $u = \sqrt{(2\log(2/\delta))/W_{\mathrm{eff}}}$. $\qquad\square$

**Statement (Theorem 2).** Let $\Delta = \widehat{F}_B^{\mathrm{old}} - \widehat{F}_B^{\mathrm{new}}$ on a batch $B$ of effective size $|B|_{\mathrm{eff}}$. Assume $\Delta - \mathbb{E}[\Delta]$ is sub–exponential with proxy $(\nu, b)$: $\mathbb{E}[\exp(\lambda(\Delta - \mathbb{E}\Delta))] \leq \exp(\frac{\lambda^2\nu^2}{2})$ for $|\lambda| \leq 1/b$. Then for any $\varepsilon > 0$ and $\delta \in (0, 1)$,

$$|B|_{\mathrm{eff}} \geq \frac{2\nu^2}{\varepsilon^2}\log\frac{1}{\delta} + \frac{2b}{3\varepsilon}\log\frac{1}{\delta} \implies \Pr(\Delta \geq \varepsilon) \geq 1 - \delta.$$

**Proof of Theorem 2.** **Step 1 (Bernstein tail bound).** For sub–exponential $X := \Delta - \mathbb{E}[\Delta]$ with proxy $(\nu, b)$,

$$\Pr(X \leq -t) \ \leq \ \exp\Big( - \frac{t^2}{2\nu^2 + 2bt/3} \Big), \qquad t > 0.$$

This is the standard one-sided Bernstein inequality derived from the MGF condition.

**Step 2 (From mean gain to high-probability gain).** We seek $\Pr(\Delta \geq \varepsilon) \geq 1 - \delta$. Write

$$\Pr(\Delta < \varepsilon) = \Pr\big(\Delta - \mathbb{E}[\Delta] < \varepsilon - \mathbb{E}[\Delta]\big) = \Pr\big(X < -t\big), \quad t := \mathbb{E}[\Delta] - \varepsilon.$$

If the *expected* gain satisfies $\mathbb{E}[\Delta] \geq \varepsilon$ then $t \geq 0$ and

$$\Pr(\Delta < \varepsilon) \leq \exp\Big( - \frac{t^2}{2\nu^2 + 2bt/3} \Big) \ \leq \ \exp\Big( - \frac{\varepsilon^2}{2\nu^2 + 2b\varepsilon/3} \Big),$$

using monotonicity of $t \mapsto t^2/(2\nu^2 + 2bt/3)$ on $t \geq 0$. To make this $\leq \delta$ it suffices that

$$\frac{\varepsilon^2}{2\nu^2 + 2b\varepsilon/3} \ \geq \ \log(1/\delta) \quad \Longleftrightarrow \quad 2\nu^2 \log(1/\delta) + \frac{2b}{3}\varepsilon \log(1/\delta) \ \leq \ \varepsilon^2.$$

Rearranging gives the stated sufficient condition on $|B|_{\text{eff}}$ once we note that the proxies $\nu^2, b$ scale as $1/|B|_{\text{eff}}$ for averages. Equivalently, write $\nu^2 = \tilde{\nu}^2/|B|_{\text{eff}}$ and $b = \tilde{b}/|B|_{\text{eff}}$ for single-sample proxies $(\tilde{\nu}^2, \tilde{b})$, then solve for $|B|_{\text{eff}}$:

$$|B|_{\text{eff}} \ \geq \ \frac{2\tilde{\nu}^2}{\varepsilon^2} \log\frac{1}{\delta} + \frac{2\tilde{b}}{3\,\varepsilon} \log\frac{1}{\delta}.$$

We re-denote $(\tilde{\nu}^2, \tilde{b})$ as $(\nu^2, b)$ in the theorem statement. $\qquad\square$

**Corollary 4** (Choosing $W$ and $|B|$)**.** *Pick $W$ so that the warm–start generalization error is at most $\eta$: $W_{\text{eff}} \geq 2\log(2/\delta)/\eta^2$. Then choose $|B|$ via Theorem 2 for target improvement $\varepsilon$ and confidence $1 - \delta$. The total decision error (from warm–start approximation and batch detection) is bounded by $\eta + \varepsilon$ at confidence $1 - 2\delta$.*

**Proof of Corollary 4.** Lemma 1 ensures $|\widehat{F}_W - F^\star| \leq \eta$ with prob. $\geq 1 - \delta$ when $W_{\text{eff}} \geq 2\log(2/\delta)/\eta^2$. Theorem 2 ensures $\Pr(\Delta \geq \varepsilon) \geq 1 - \delta$ when the batch bound holds. By a union bound, the probability that both events hold is at least $1 - 2\delta$. If both hold, the total decision error (warm–start approximation plus detection slack) is at most $\eta + \varepsilon$. $\qquad\square$

**How Many Soft–Prompt Candidates? Coverage and Suboptimality**

**Statement (Theorem 3).** Let the active local subspace be $r_{\text{act}}$–dimensional with unit sphere $\mathbb{S}^{r_{\text{act}}-1}$. Fix $\boldsymbol{v}_1$ and draw i.i.d. $\boldsymbol{u}_1, \ldots, \boldsymbol{u}_M \sim \text{Unif}(\mathbb{S}^{r_{\text{act}}-1})$. Then for any $\theta \in (0, \pi/2]$,

$$\Pr\Big( \min_{m \leq M} \angle(\boldsymbol{u}_m, \boldsymbol{v}_1) \leq \theta \Big) \ \geq \ 1 - \Big( 1 - (\sin\theta)^{r_{\text{act}}-1} \Big)^M.$$

**Proof of Theorem 3.** **Step 1 (Exact cap probability).** WLOG set $\boldsymbol{v}_1$ as the north pole. For $\boldsymbol{U} \sim \text{Unif}(\mathbb{S}^{r_{\text{act}}-1})$, the random variable $T = \langle \boldsymbol{U}, \boldsymbol{v}_1 \rangle$ has density $f_T(t) \propto (1 - t^2)^{(r_{\text{act}}-3)/2}$ on $t \in [-1, 1]$. Hence $\Pr(\angle(\boldsymbol{U}, \boldsymbol{v}_1) \leq \theta) = \Pr(T \geq \cos\theta) = I_{\sin^2\theta}(\frac{r_{\text{act}}-1}{2}, \frac{1}{2})$.

**Step 2 (Lower bound).** For $\theta \in (0, \pi/2]$, $\sin\theta \in (0, 1]$ and we have the elementary bound

$$I_{\sin^2\theta}\Big( \frac{r_{\text{act}}-1}{2}, \frac{1}{2} \Big) \ \geq \ (\sin\theta)^{r_{\text{act}}-1}.$$

(Proof: $I_x(a, b) = \frac{1}{B(a,b)} \int_0^x t^{a-1}(1 - t)^{b-1} dt \geq \frac{1}{B(a,b)} \int_0^x t^{a-1} dt = \frac{x^a}{a\,B(a,b)} \geq x^a$ since $a\,B(a, b) \leq 1$ for $a \geq 1/2, b \geq 1/2$; let $x = \sin^2\theta$ and $a = (r_{\text{act}} - 1)/2$.)

**Step 3 (Independence across $M$ samples).** Thus, for one sample, $p_\theta := \Pr(\angle(\boldsymbol{U}, \boldsymbol{v}_1) \leq \theta) \geq (\sin\theta)^{r_{\text{act}}-1}$. The probability *none* among $M$ falls in the cap is $(1 - p_\theta)^M \leq (1 - (\sin\theta)^{r_{\text{act}}-1})^M$. Therefore

$$\Pr\Big( \min_{m \leq M} \angle(\boldsymbol{u}_m, \boldsymbol{v}_1) \leq \theta \Big) = 1 - (1 - p_\theta)^M \ \geq \ 1 - \Big( 1 - (\sin\theta)^{r_{\text{act}}-1} \Big)^M.$$

$\qquad\square$

**Statement (Lemma 2).** If $\angle(\hat{\boldsymbol{u}}, \boldsymbol{v}_1) \leq \theta$ and $\mathbf{H}_T \succeq \mathbf{0}$ with top eigenpair $(\lambda_1, \boldsymbol{v}_1)$, then

$$\hat{\boldsymbol{u}}^\top \mathbf{H}_T \hat{\boldsymbol{u}} \geq \lambda_1 \cos^2\theta \quad \Rightarrow \quad \lambda_1 - \hat{\boldsymbol{u}}^\top \mathbf{H}_T \hat{\boldsymbol{u}} \leq \lambda_1 \sin^2\theta.$$

**Proof of Lemma 2.** Decompose $\hat{\boldsymbol{u}} = \cos\theta\,\boldsymbol{v}_1 + \sin\theta\,\boldsymbol{w}$ with $\|\boldsymbol{w}\|_2 = 1$ and $\boldsymbol{w} \perp \boldsymbol{v}_1$. Then

$$\hat{\boldsymbol{u}}^\top \mathbf{H}_T \hat{\boldsymbol{u}} = \lambda_1 \cos^2\theta + \sum_{j \geq 2} \lambda_j \langle \boldsymbol{v}_j, \boldsymbol{w} \rangle^2 \sin^2\theta \geq \lambda_1 \cos^2\theta,$$

since $\lambda_j \geq 0$. Subtract from $\lambda_1$ to obtain the residual bound. □

**Proof of Corollary 3.** We require $1 - (1 - (\sin\theta)^{r_{\mathrm{act}}-1})^M \geq 1 - \delta$. Equivalently $(1 - (\sin\theta)^{r_{\mathrm{act}}-1})^M \leq \delta$, giving

$$M \geq \frac{\log(1/\delta)}{\log\left((1 - (\sin\theta)^{r_{\mathrm{act}}-1})^{-1}\right)}.$$

□

# E    ADDITIONAL EXPERIMENT RESULTS

In this section, we present additional experiment results to complement the main paper.

## E.1    RESPONSE QUALITY FOR QWEN

To further answer how well SOMA performs against baselines with different backbone models, we replicate our full evaluation on a second model family (Qwen). Beyond the LLaMA results in Table 1, Table 3 reports the similarity percentage of each method's responses to the original model across six datasets when the original is Qwen-3-8B and the surrogate is Qwen-3-0.6B. The experimental setting, judge, and data splits are identical to those used for LLaMA. Overall, SOMA remains the top method across datasets. In every dataset, SOMA achieves the highest similarity to the original model, mirroring the pattern observed for LLaMA. This reinforces that our approach generalizes across architectures and tokenizer vocabularies. One thing to notice is that, compared with Table 1, similarities are generally lower and per–dataset standard deviations are larger. This gap is expected for three reasons: (i) the capacity gap between Qwen-3-8B and Qwen-3-0.6B is substantially larger than that between LLaMA-3.1-70B and LLaMA-2-7B, making the imitation task intrinsically harder; (ii) differences in pretraining and instruction alignment lead to stronger style and reasoning mismatches that a small LoRA has to compensate for; and (iii) tokenization and calibration differences (e.g., subword boundaries and logit scaling) introduce additional noise in the token-level comparison used by the judge, which inflates variance. But taken together, Table 1 and Table 3 show that SOMA provides a consistent quality boost over competitive baselines while preserving the token/throughput advantages of using a small surrogate. In the more challenging Qwen setting, the absolute ceiling is lower, but the relative benefit of SOMA is as strong or stronger, indicating that our local adaptation is especially valuable when the small–large gap is wide.

## E.2    DETAILS OF EFFICIENCY RESULTS

To proxy the API cost for different datasets, we measured the average tokens per dialogue for all methods on six datasets in each family. As shown in Figure 6 and Figure 7, SOMA consistently uses the fewest tokens per dialogue. This comes from switching to the adapted small model after a short warm start, so we stop re-encoding the long head and drop soft prompts at service time. As a result, SOMA avoids the large token budgets of Original and History-Prefix, and is leaner than Surrogate/History-FT because later turns are no longer as SOMA compressed the full history for later turns. The effect is most visible on Qwen, where early turns are proportionally heavier and the savings from early truncation are larger.

To examine the runtime when using different methods on real-world datasets, we tested the throughput (tokens/sec) for all methods on the same six datasets. The results in Figure 8 and Figure 9 show SOMA matches or exceeds the speed of Surrogate/History-FT and clearly outperforms

Table 3: Similarity percentage to the original model across six datasets for the Qwen family.

| | ShareGPT | ReMeDi | Craigslist | Multi-Char | MATH | MT-Bench | Avg |
|---|---|---|---|---|---|---|---|
| Surrogate | $50.9 \pm 1.21$ | $63.5 \pm 2.87$ | $48.2 \pm 3.40$ | $44.7 \pm 3.54$ | $42.6 \pm 1.70$ | $40.4 \pm 0.81$ | $48.4 \pm 8.31$ |
| History-Prefix | $70.5 \pm 1.15$ | $75.7 \pm 2.01$ | $63.0 \pm 2.23$ | $65.5 \pm 0.91$ | $51.3 \pm 2.45$ | $53.6 \pm 2.32$ | $63.3 \pm 9.47$ |
| History-FT | $76.4 \pm 1.68$ | $79.2 \pm 2.61$ | $74.5 \pm 3.56$ | $63.2 \pm 1.74$ | $65.9 \pm 1.70$ | $62.7 \pm 0.72$ | $70.3 \pm 7.23$ |
| LLMLingua-2 | $68.9 \pm 1.32$ | $74.1 \pm 2.14$ | $61.2 \pm 2.67$ | $63.4 \pm 1.18$ | $49.8 \pm 2.03$ | $51.7 \pm 1.45$ | $61.5 \pm 7.92$ |
| RouteLLM | $79.6 \pm 1.04$ | $81.9 \pm 1.36$ | $75.1 \pm 2.91$ | $72.8 \pm 1.62$ | $67.3 \pm 1.21$ | $68.0 \pm 0.96$ | $74.1 \pm 5.43$ |
| **SOMA** | $\mathbf{81.0 \pm 0.93}$ | $\mathbf{83.2 \pm 1.21}$ | $\mathbf{76.4 \pm 3.85}$ | $\mathbf{74.2 \pm 2.53}$ | $\mathbf{68.7 \pm 1.28}$ | $\mathbf{69.2 \pm 1.08}$ | $\mathbf{75.5 \pm 5.97}$ |

Original/History-Prefix. The gain is explained by two design choices validated in the experiment: (i) after switching, responses are produced by the small model; and (ii) inputs are shorter because the long head is not reprocessed every turn. Thus, SOMA delivers surrogate-level throughput while maintaining high similarity to the original model's outputs.

### E.3    Ablation Study for Qwen

To test whether the findings in Section 5.1 that each component of SOMA performs well can generalize beyond LLaMA, we repeated the ablations on the Qwen family (Figure 10). Same as the LLaMA results, we evaluate SOMA, SOMA w/o ADL (removing the anti-degeneration entropy regularizer), and SOMA w/o ExpW+ADL (removing both the expectation-weighted term and ADL). We find the same pattern: the full SOMA achieves the highest similarity on every dataset; dropping ADL consistently lowers scores, showing that preserving tail entropy during prompt mining prevents probability collapse and stabilizes learning; removing both components hurts the most, especially on harder sets, confirming that the expectation-weighted term is crucial for penalizing distribution-level semantic alignment that token-level unlikelihood misses. Overall, the Qwen ablations echo the LLaMA study and demonstrate that both ADL and expectation-weighting are necessary for consistent gains.

### E.4    Case study: SOMA delivers broad and balanced gains across abilities

To examine how SOMA improves specific capabilities compared with baselines, we follow the MT-Bench-101 benchmark (Bai et al., 2024) and run a fine-grained evaluation on the LLaMA family. We use the same LLM judge as in the main experiments and the benchmark's prompts; scores are on a 0–10 scale, where higher is better. As shown in Figure 11, SOMA delivers the largest gains on the harder skills—reasoning and questioning—precisely where the original model most exceeds the surrogate. At the same time, SOMA lifts memory, understanding, rephrasing, and interference, indicating that the learned local adaptation not only reduces the high-level reasoning gap but also strengthens general dialog competence. Overall, SOMA yields a consistently stronger and more balanced ability profile than History-Prefix, History-FT, and the raw surrogate.

### E.5    Case study: Later switching points correlate with lower final similarity

To understand when does SOMA switch to the fine-tuned surrogate during service, we sweep the warm-start window $W \in [1, 15]$ turns and, for each dataset, measure the final SOMA similarity after switching at $W$. Figure 12 shows consistent patterns that simpler, goal-anchored dialogues (ShareGPT, ReMeDi, Craigslist) exceed high similarity after only a few turns, whereas reasoning-heavy or multi-party settings (MATH, Multi-Char) require longer context before plateauing. The curves rise steeply for small $W$ (early turns carry most supervision) and exhibit diminishing returns thereafter, indicating that late turns add little for local adaptation. Figure 13 plots the plateau score against the best $W$ per dataset and reveals a clear negative association (Pearson $r = -0.64$): tasks needing more warm-start evidence achieve lower final similarity, reflecting higher intrinsic difficulty and a larger behavior gap between the large and small models. Practically, this suggests short windows for general chats and longer windows for compositional reasoning or multi-agent dialogues. Also, once the sweep window goes beyond the point where the curve starts to flatten, adding more warm-start turns brings almost no accuracy gain but delays the switch and cuts into efficiency.

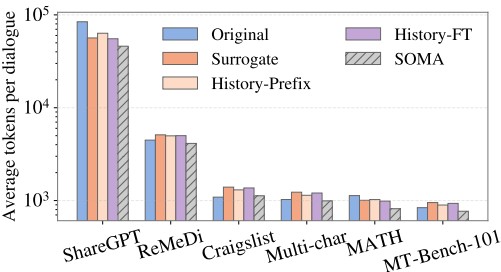

Figure 6: Average tokens per dialogue across six datasets (LLaMA family).

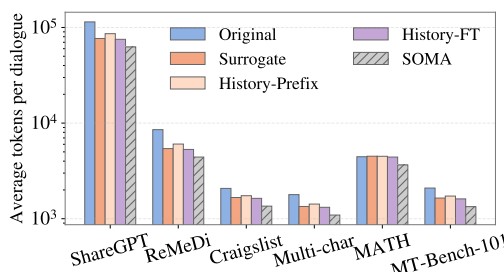

Figure 7: Average tokens per dialogue across six datasets (Qwen family).

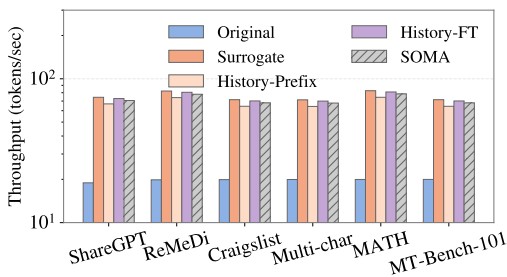

Figure 8: Throughput of using different methods on six datasets (LLaMA family)

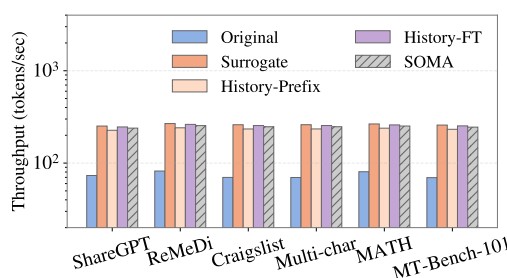

Figure 9: Throughput of using different methods on six datasets (Qwen family)

### E.6 EMPIRICAL VARIANCE CONCENTRATION AND THEORETICAL UPPER BOUND

We examine how the variance of surrogate–teacher similarity changes with the warm-start window $W$. For each dataset, we compute the empirical standard deviation across dialogues at each truncation point. As shown in Figure 14, the variance decreases steadily as $W$ grows: early turns exhibit high variability, while later turns fall into a stable, low-variance regime. This pattern supports the concentration behavior assumed in Lemma 1 and used in the switching bound of Theorem 2. Overall, the empirical variance decay is consistent with the theoretical framework of Section 4.1. Early turns provide substantial information gain, and the diminishing-variance region for larger $W$ matches the concentration assumptions underlying the switching rule. The theoretical envelope also illustrates the close alignment between SOMA's empirical behavior and its theoretical foundations.

## F LIMITATION AND FUTURE WORK

Despite advances compared with baselines, SOMA has a few limitations. The surrogate's own capacity limits how closely it can match the large model, and when the gap is very large some behaviors cannot be recovered even with fine-tuning. Soft-prompt mining also needs access to the surrogate's tokenizer and embedding space, which means the method is harder to use in strict black-box API settings. In addition, SOMA is only designed for context-dependent dialogue where context remains locally coherent; it assumes that early turns provide a smooth and stable region for learning. Sudden topic changes or strong paraphrase differences can weaken the mined directions and make switching less reliable. The method also adds a small probe cost at the beginning of a conversation, and our evaluation uses an LLM judge, which may carry minor bias.

Future work includes using better drift detectors, exploring approximate mining that works with limited internal access, and improving robustness when the surrogate is much smaller than the teacher. Another direction is to learn multiple local regions for multi-topic dialogues, develop privacy-preserving mining procedures, and amortize soft-prompt search across sessions. Extending SOMA to multi-modal models is also a promising next step.

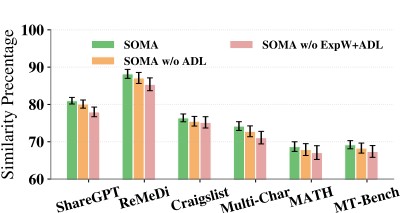

Figure 10: Ablation Studies on Qwen family

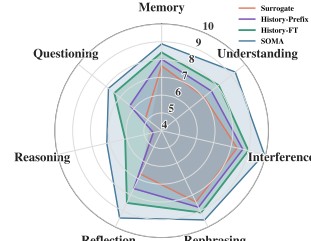

Figure 11: Performance of different methods across various ability dimensions (LLaMA).

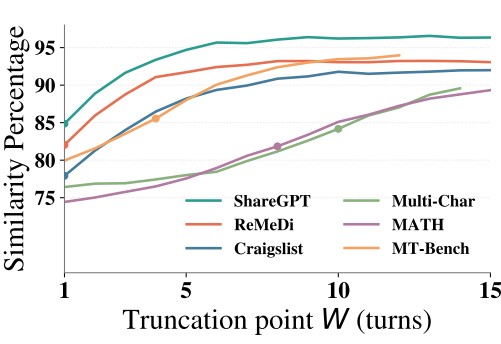

Figure 12: Average turns needed before switching on each dataset.

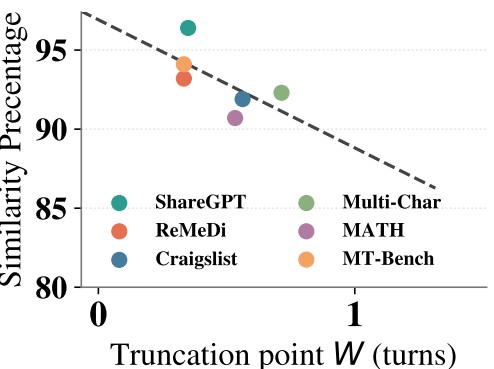

Figure 13: Average turns needed before switching is negatively correlated with performance.

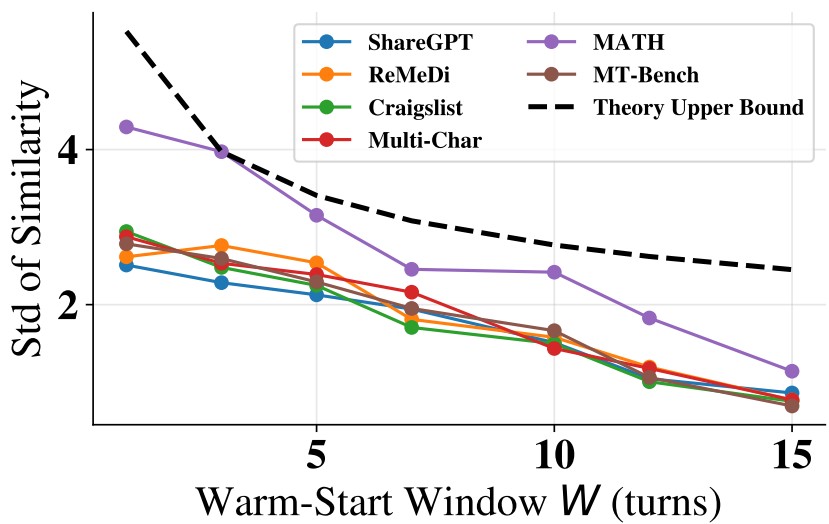

Figure 14: Variance of Similarity vs. Warm-Start Window.

