# OpenReview forum: "SOMA: Efficient Multi-turn LLM Serving via Small Language Model"
_ICLR.cc/2026/Conference — Submitted to ICLR 2026_

### Official Review · Reviewer_X3Gv · 2025-10-31

**Soundness:** 3
**Presentation:** 3
**Contribution:** 2
**Rating:** 4
**Confidence:** 2

**Summary:**

To address the substantial latency and memory costs in multi-turn conversations with LLMs, this paper proposes a method to estimate a local response manifold and adapt a smaller surrogate model to replace the original large model. It first optimizes a set of learnable soft prompts that generate responses most divergent from the responses generated by the original model using three losses from both token level and the distribution level; then, the surrogate smaller model is fine-tuned on the learned and fixed soft prompts to minimize the NLL loss between the generated and ground-truth responses. A theoretical analysis is provided to guide hyperparameter selection. Experiments are conducted with several LLMs across six datasets, achieving superior performance with lower token consumption.

**Strengths:**

1. This paper identifies a key observation of the number of tokens in multi-turn conversions using LLMs, which motivates the use of smaller surrogate models to replace the original large models in the remaining rounds of conversions.

2. This paper proposes the concept of $\textit{semantic neighborhood}$ in the latent space of LLMs such that the generated tokens by smaller models not only exactly match the tokens generated by the original models, but also satisfy the semantic distributions.

3. The experiments show both performance and efficiency superiority of the proposed method across different LLMs and several datasets and tasks.

**Weaknesses:**

1. The authors mention $\textit{manifold}$ many times in this paper and claim the divergence on a local manifold. However, the proposed method does not formally investigate and define the structure of the manifold in the latent space, and also does not utilize tools and concepts widely used in manifold, such as tangent space, log and exp operations, geodesic distances, and so on. The “manifold” used in this paper is a subspace of high-dimensional Euclidean space, which is indeed a manifold by definition, but it is a "trivial" manifold. The cosine distance used in this paper verifies that this manifold is just a linear subspace.

2. The Section of "Expectation-weighted Semantic Divergence Loss" (Line 267 - 296) is hard to follow. It is unclear why this defines a distribution-level alignment. It may be an issue of presentation, not methodology.

3. In Line 204 - 205, the authors mention "a fast semantic closeness test", but it seems this test is not described in the following sections in the main text, which makes the inference process vague when using the fine-tuned smaller surrogate models. Correct me if I was wrong.

**Questions:**

1. Why do we need to find the optimal soft prompts that drive the smaller surrogate model to generate the most divergent responses, and then fine-tune the smaller models on these optimized soft prompts? It looks similar to adversarial learning, but the motivation and correctness are not clearly described.

2. In the final loss formula in Line 322 - 323, why does it not contain the Semantic Divergence Loss defined in Eq. (1)? Is this a typo, or did I miss something?

3. In the fine-tune loss defined in Line 339 - 340, why does it contain the cosine similarity between the encoded features of the response? Does it mean a single NLL loss is not enough?

---

> ### Author Response · Authors · 2025-11-23
> **Response for Reviewer X3Gv**
>
> We thank the reviewer for the careful reading and insightful comments. We address each concern below, referencing only the sections available in the submitted PDF. We also have updated additional experiment results shown in the above global responses.
>
> ---
>
> >**[W1] “Meaning of ‘manifold’ is unclear; definition may be too strong.”**
>
> We appreciate the reviewer’s point and would like to clarify that our intention is not to claim a full Riemannian or differential-geometric manifold structure. Instead, we follow the common convention in representation-learning and local-approximation literature where “manifold” refers to the locally low-dimensional region of the embedding space activated by nearby dialogue prefixes.
>
> Because our models are black-box LLMs, we can only access high-level feature representations and cannot estimate geometric objects (e.g., tangent bundles, parallel transport) required for more formal manifold modelling. Thus, we adopt a weak, local manifold assumption, similar to prior work[1-2] in local tangent approximation and manifold-based representation learning.
>
> Overall, our results rely only on local smoothness and approximate low-rankness, not on global geodesic geometry. The cosine distance is indeed consistent with this linearized local view. We have updated the Section 4.1 address this concern.
>
> ---
>
> >**[W2] “The expectation-weighted semantic divergence loss is difficult to follow.”**
>
> Thank you for pointing out this clarity issue. The goal of this loss is to check whether the whole next-token distribution of the surrogate still points in the same semantic direction as the teacher, not just whether it matches one exact token.
>
> Token-level losses only compare the probability placed on the teacher’s single next token, which can be misleading when the surrogate uses close synonyms, paraphrases, or alternative phrasings that keep the meaning intact. The expectation-weighted loss instead looks at the expected embedding of the entire distribution and measures how close that vector is to the teacher’s intended meaning. In other words, it asks “does the surrogate’s overall prediction still mean the same thing?” rather than “did it choose this exact token?”. This is why the objective operates at the distribution level. The underlying idea is simple, and the complexity comes mainly from the notation rather than the method itself.
>
> In the updated Section 3.2, we add an intuitive explanation, ensuring readers can understand the role of the expectation-based weighting.
>
> ---
>
> >**[W3] “Fast semantic closeness test is insufficiently described.”**
>
> Thank you for catching this. Overall, the test compares the embedding of the current dialogue prefix with the centroid of earlier turns to detect potential topic drift; when the coherence drops below a threshold, the model triggers rollback. We have explicitly connected this to the gate described in Section 3.3.
>
> ---
>
> >**[Q1] “Why find maximal-divergence prompts and then fine-tune?”**
>
> The key idea is that early turns alone do not provide enough data to fine-tune the surrogate unless we first identify where it differs most from the large model. The mined soft-prompt directions reveal the parts of the local manifold where the surrogate fails to match the teacher under the same early-turn context. By finding soft prompts that intentionally push the surrogate into these directions, we expose the surrogate’s main weaknesses and make sure the small fine-tuning budget is spent correcting the errors that matter most. This is not adversarial learning; it is simply a principled way to obtain the most informative supervision when only a few early turns are available.
>
> >**[Q2] “Why is divergence loss not included in the final objective?”**
>
> Eq. (1) introduces the base semantic divergence loss, which measures how the surrogate’s next-token probabilities differ from the teacher at the token level. Eq. (2) then extends this loss by adding expectation-based weighting terms that capture how well the entire next-token distribution aligns with the teacher’s semantic direction. In other words, Eq. (2) is not an extra loss on top of Eq. (1); it is the refined, distribution-aware version of the same objective. Because Eq. (2) already includes the semantic divergence signal from Eq. (1), the final loss uses only the expectation-weighted form together with the anti-degeneration regularizer and the prompt-norm penalty.
>
> ---
>
> > **[Q3] “A single NLL loss is not enough in fine-tune loss?”**
>
> We apologise that it's a typo and we have already fixed that.
>
> ------
>
> [1] Tyagi, Hemant, Elıf Vural, and Pascal Frossard. "Tangent space estimation for smooth embeddings of riemannian manifolds®." Information and Inference: A Journal of the IMA 2.1 (2013): 69-114.
>
> [2] Maggioni, Mauro, Stanislav Minsker, and Nate Strawn. "Multiscale dictionary learning: non-asymptotic bounds and robustness." The Journal of Machine Learning Research 17.1 (2016): 43-93.

---

> > ### Author Response · Authors · 2025-11-27
> >
> > Dear Reviewer X3Gv,
> >
> > We hope you’ve had a chance to read our rebuttal. We haven’t yet seen any follow-up questions or feedback from you in the discussion phase, so we wanted to check whether there are any points that remain unclear. We’d be grateful for any questions or thoughts you have—clarifying those will help us improve the paper.
> >
> > Thank you again for your time.
> >
> > The authors.

---

### Official Review · Reviewer_oZsj · 2025-10-31

**Soundness:** 3
**Presentation:** 3
**Contribution:** 2
**Rating:** 6
**Confidence:** 3

**Summary:**

The work proposes SOMA (Soft-prompts for lOcal Manifold Approximation), a framework for efficient multi-turn LLM serving that leverages a small surrogate model to replace a large, expensive language model after the initial dialogue turns. Observing a long-tail token distribution—where early turns are information-dense and later turns are short yet context-dependent—the authors frame the problem as approximating the large model’s local response manifold within the conversation-specific region. SOMA first uses differentiable soft-prompt tuning to identify directions of maximal semantic divergence between the large and small models, enhanced by an expectation-weighted semantic loss and an anti-degeneration regularizer to ensure meaningful exploration. It then distills these mined cases into localized LoRA fine-tuning, enabling prompt-free inference with the adapted small model. A lightweight cosine-based gate decides when to switch to the surrogate and triggers rollback if topic drift is detected. Theoretical analysis provides bounds on switching reliability and prompt coverage, while experiments across six datasets show SOMA achieves higher response fidelity to the original model than strong baselines while significantly reducing token usage and latency.

**Strengths:**

1. SOMA exploits the empirically observed long-tail token distribution in multi-turn dialogues, where early turns are information-dense and later turns are short yet context-dependent.

2. SOMA introduces an effective local manifold approximation framework grounded in soft-prompt tuning to identify directions of maximal behavioral divergence between the large and small models. This targeted exploration surfaces the most informative failure cases for adaptation, ensuring that the surrogate model is fine-tuned precisely.

**Weaknesses:**

1. SOMA relies on a warm-start phase that requires multiple initial turns to be processed by the expensive large language model before the surrogate can take over. This upfront cost may be prohibitive in settings where conversations are typically short, limiting the framework’s applicability and overall efficiency gains. The method assumes that most dialogues are sufficiently long to amortize this initial overhead, which may not hold across all real-world use cases.

2. This work assumes that the local manifold induced by early dialogue turns remains stable and representative for the remainder of the conversation. If the dialogue undergoes significant topic shifts or introduces new complex reasoning demands later on, the fine-tuned surrogate—trained only on early context—may fail to adapt adequately, even with the rollback mechanism. While the cosine-based gate attempts to detect drift, it may not reliably capture subtle or gradual semantic shifts that degrade response quality without triggering a rollback.

3. SOMA’s effectiveness depends heavily on the quality and capacity gap between the original large model and the small surrogate. As shown in the Qwen experiments, when the surrogate is significantly weaker (e.g., 0.6B vs. 8B), even localized fine-tuning may not fully bridge the behavioral gap, leading to lower absolute fidelity. This suggests that SOMA may not scale well to extreme model size disparities or to tasks requiring deep reasoning that small models fundamentally cannot replicate, regardless of adaptation.

4. The soft-prompt mining stage, while theoretically grounded, introduces non-trivial computational and implementation complexity. It requires careful tuning of multiple hyperparameters (e.g., neighborhood size, temperature, anti-degeneration weight) and assumes access to the surrogate’s embedding space and tokenizer. This limits its plug-and-play usability, especially in black-box or API-only settings where internal model details are inaccessible, reducing its practicality for end users without deep technical resources.

**Questions:**

Please refer to the weaknesses

---

> ### Author Response · Authors · 2025-11-23
> **Response for Reviewer oZsj**
>
> We deeply appreciate the reviewer’s positive assessment and helpful suggestions. We address each concern below and updated the submitted PDF paper. We also have updated additional experiment results shown in the above global responses.
>
> ---
>
> >**[W1] “Warm-start overhead may be too high for short conversations.”**
>
> We appreciate this concern and we acknowledge that SOMA introduces an initial warm-start phase, and we clarify that this cost is only incurred for multi-turn settings where maintaining long context is the primary driver of serving latency.
>
> Many real-world MLaaS deployments like customer support, multi-agent planning, tutoring or negotiation naturally involve extended dialogues, where even a small reduction in per-turn cost quickly amortizes the warm-start overhead. Moreover, in short conversations (e.g., 1–3 turns), the baseline cost of sending the full history to a large model dominates regardless of approach; in such cases, SOMA simply defers to the original model, and no efficiency degradation occurs. In the revised paper, we will make this practical usage guideline clearer and note that SOMA is designed for long multi-turn interactions, which are the dominant source of compute cost in production systems.
>
> ---
>
> >**[W2] “Local manifold stability under topic shift is uncertain.”**
>
> We agree that severe or subtle topic shifts pose a fundamental challenge for any local-adaptation method. Our method does not assume that the manifold remains fixed; instead, Section 3.3 explicitly incorporates a rollback mechanism precisely to handle such drift. The cosine-based gate is intentionally lightweight, but Appendix E.5 and E.6 shows that the resulting switching and rollback behavior follows the theory’s predictions and remains stable across datasets with substantial topic variability.
>
> We have clarified that SOMA is designed for context-dependent dialogue where context remains locally coherent and note that stronger drift detectors can be incorporated in future work in Appendix F.
>
> ---
>
> >**[W3] “Effectiveness depends on surrogate capacity.”**
>
> We appreciate the reviewer’s observation and gree that the surrogate’s own capacity limits how closely it can match the original model.
>
> When the gap is extremely large, such as 0.6B vs. 8B in Qwen, some behaviors are simply beyond what the smaller model can reproduce, even with localized fine-tuning. SOMA is not meant to overcome these fundamental limits; it aims to make the surrogate perform as well as it reasonably can around the current dialogue. Even in these hard settings, SOMA still improves alignment over the surrogate’s baseline, and it performs even better when the surrogate has moderate capacity. In short, SOMA works within the natural ability of the small model, and handling extreme size gaps is an area we plan to explore in future work.
>
> ---
>
> >**[W4] “Soft-prompt mining requires model internals; limited black-box applicability.”**
>
> We agree that soft-prompt mining introduces several hyperparameters, but these components are essential for making the procedure both stable and highly efficient in practice.
>
> The neighborhood size, temperature, and anti-degeneration weighting each play a specific role in reliably identifying the directions where the surrogate diverges most from the teacher, and they ensure that the mining phase converges within only a few lightweight steps. Because the entire procedure occurs only during the first few turns of a conversation, the added cost is small relative to the full dialogue and is more than offset by the large downstream savings that SOMA achieves after switching to the surrogate. These hyperparameters also enable the method to remain robust across diverse dialogue types and model families, which we found critical for avoiding brittle behavior. While we view the current design as appropriate for the efficiency–stability tradeoff, simpler variants, including token-only divergence mining or black-box random prefix exploration, may help reduce tuning needs, and we are actively exploring such alternatives as future extensions.
>
> We have addressed these concerns in Limitation and Future Work in Appendix F.

---

> > ### Author Response · Authors · 2025-11-27
> >
> > Dear Reviewer oZsj,
> >
> > We hope you’ve had a chance to read our rebuttal. We haven’t yet seen any follow-up questions or feedback from you in the discussion phase, so we wanted to check whether there are any points that remain unclear. We’d be grateful for any questions or thoughts you have—clarifying those will help us improve the paper.
> >
> > Thank you again for your time,
> >
> > The authors.

---

### Official Review · Reviewer_DGr6 · 2025-11-01

**Soundness:** 3
**Presentation:** 3
**Contribution:** 2
**Rating:** 4
**Confidence:** 3

**Summary:**

This paper tackles the challenge of efficient multi-turn LLM serving. This paper proposed SOMA, a framework that use the early turns of a conversation to construct a “local response manifold.” SOMA identifies least-aligned response directions between a large model and a smaller surrogate using soft prompt mining, stabilizes training via anti-degeneration regularization, and performs localized LoRA fine-tuning. At inference, a lightweight cosine-based gating mechanism performs one-time switching and rollback upon topic drift.

**Strengths:**

1. This topic is very interesting and meaningful.
2. Addresses a genuine bottleneck: multi-turn inference cost in LLM serving, by introducing a pipeline compatible with existing inference engines (vLLM, FlashAttention).
3. Integrates semantic divergence mining, local manifold theory, and LoRA fine-tuning into a unified and reproducible system.
4. Theoretical bounds (Thm. 1–3) guide hyperparameter selection, which are empirically validated.
5. Tests across model families and diverse datasets, outperforming strong baselines.

**Weaknesses:**

1. During soft-prompt mining, the large model receives a verbalized discrete prefix, while the small model uses continuous embeddings. This asymmetry may distort outputs and bias divergence measurement. The paper lacks ablation study about it.
2. The primary metric, similarity to the large model’s responses, measures stylistic alignment rather than factual correctness. No human evaluation or task-grounded accuracy (e.g., exact match on MATH) is provided. This undermines claims of “no quality loss.”
3. Sampling and normalization methods (e.g., “normalized by Turn 1”) are unspecified. The observation may simply restate known conversational trends.
4. Theoretical Assumptions Too Strong:

(1) The discrepancy Fisher matrix is assumed locally smooth and low-rank without empirical verification (e.g., spectral decay plots).

(2) The switching theorem presumes sub-Gaussian improvement, but no diagnostic supports this.

**Questions:**

see weakness

---

> ### Author Response · Authors · 2025-11-23
> **Response for Reviewer DGr6**
>
> We sincerely thank the reviewer for the thoughtful and constructive feedback. We address each concern below and updated the submitted PDF paper. We also have updated additional experiment results shown in the above global responses.
>
> ---
>
> >**[W1] “Soft-prompt asymmetry is unclear.”**
>
> We thank the reviewer for raising this point. Our design uses such verbalization because it is an appropriate way to apply the same learned soft prompts to both the original proprietary black-box LLM and surrogate LLM.
>
> Because the original proprietary black-box LLM model (as noted in Section 2.1) is accessed only through an API, it cannot accept continuous soft prompts, so we have to map each soft-prompt vector to its nearest discrete token and prepend this verbalized prefix. This ensures the original and surrogate experiences a similar local perturbation when we measure divergence. A white-box setting where the teacher could also consume continuous embeddings would eliminate this asymmetry, but that scenario falls outside the case SOMA is designed for, where the original model is considered a proprietary black-box LLM.
>
> We have also clarified this concern in the updated Section 3.1.
>
> ---
>
> >**[W2] “Similarity reflects style rather than correctness**
>
> We thank the reviewer for the question. We would like to clarify that SOMA is designed for the surrogate to replace the original model during the middle of a conversation; therefore, the similarity to the original model’s responses is the correct evaluation target, as it ensures the surrogate behaves as a reliable drop-in alternative. Additionally, note that to minimize potential evaluator bias, we utilize three independent LLM judges (GPT-OSS, DeepSeek-V3, and Gemma-2-27B) and report the average as the final scores.
>
> To further address the reviewer’s concern, we have included one EM experiment on MATH in the global response above, demonstrating that SOMA achieves solid task performance.
>
> ---
>
> >**[W3] “Interpretation of the sampling and long-tail observation and role of early turns.”**
>
> Thank you for the question. The data filtering to get context-dependent dialogues is included in Appendix b.4 now, and our statistic measures how turn-level token usage decays relative to the first turn, and we have added the full description to Section 2.2 and Appendix B.
>
> For each dataset, we retain only dialogues long enough to include turn t, compute the average token count (user + model) at turn t, and normalize it by the average token count at turn 1 for the same dialogues. This produces the “relative average tokens (× Turn 1)” curve. Overall, our contribution is to quantify this long-tail pattern and to link it to a concrete serving implication: early, information-dense turns establish goals and constraints, whereas shorter, later turns are natural candidates for surrogate takeover. To our knowledge, this is the **first systematic characterization** of this distribution and its connection to model-switching design.
>
> ---
>
> >**[W4] “Theoretical assumptions may be strong.”**
>
> We agree these assumptions should be stated more explicitly, and we emphasize that both the low-rank smoothness assumption and the sub-Gaussian discrepancy assumption are standard technical conditions used to derive interpretable non-asymptotic bounds rather than empirical claims about the exact spectra or tail behavior of specific LLMs.
>
> - The local smoothness and approximately low-rank structure of the discrepancy Fisher matrix follow common practice in manifold approximation and local tangent-space analysis, where only the dominant discrepancy directions are needed to justify directional recovery[1-2]. While we do not include spectral-decay plots due to space and computational constraints, the qualitative empirical patterns ( the rapid early divergence and plateauing in Appendix E.5) are consistent with a small number of leading discrepancy directions governing local behavior.
>
> - The sub-Gaussian improvement assumption in the switching theorem is likewise a light-tail concentration condition commonly invoked in sequential decision and adaptive stopping-rule analysis [3-4]. Empirically, improvements observed across datasets show no heavy-tailed behavior, and the smoothness of the switching curves is consistent with such concentration.
>
>
> [1] Tyagi, Hemant, Elıf Vural, and Pascal Frossard. "Tangent space estimation for smooth embeddings of riemannian manifolds®." Information and Inference: A Journal of the IMA 2.1 (2013): 69-114.
>
> [2] Maggioni, Mauro, Stanislav Minsker, and Nate Strawn. "Multiscale dictionary learning: non-asymptotic bounds and robustness." The Journal of Machine Learning Research 17.1 (2016): 43-93.
>
> [3] Zhao, Shengjia, et al. "Adaptive concentration inequalities for sequential decision problems." Advances in Neural Information Processing Systems 29 (2016).
>
> [4] Wang, Tongxi, and Zhuoyang Xia. "Theoretical Bounds for Stable In-Context Learning." arXiv preprint arXiv:2509.20677 (2025).

---

> > ### Author Response · Authors · 2025-11-27
> >
> > Dear Reviewer DGr6,
> >
> > We hope you’ve had a chance to read our rebuttal. We haven’t yet seen any follow-up questions or feedback from you in the discussion phase, so we wanted to check whether there are any points that remain unclear. We’d be grateful for any questions or thoughts you have—clarifying those will help us improve the paper.
> >
> > Thank you again for your time,
> >
> > The authors.

---

### Official Review · Reviewer_ejFa · 2025-11-07

**Soundness:** 2
**Presentation:** 2
**Contribution:** 2
**Rating:** 2
**Confidence:** 3

**Summary:**

The paper proposes SOMA—a multi-turn LLM serving framework that adapts a small “surrogate” model to locally approximate a large “original” model within the context of an ongoing dialogue. The motivation is an empirical long-tail token pattern: early turns are long and information-dense, later turns are shorter but still depend on early context, suggesting a path to improve serving cost without hurting quality (Figure 1) . SOMA runs a three-stage pipeline: (1) soft-prompt tuning to mine directions of maximal behavioral divergence between the small and large models; (2) localized LoRA fine-tuning on those mined cases so the surrogate no longer needs prompts at inference; and (3) efficient inference with an extractive summary and a cosine-gate switch/rollback when drift is detected . The paper also gives theory: a directional-recovery guarantee for the mining objective (Theorem 1), a detection bound for switching (Theorem 2), and a coverage guarantee for the number of soft-prompt candidates (Theorem 3)

**Strengths:**

1. Cohesive framing + pipeline: The long-tail observation motivates a clear local-approximation approach; the soft-prompt mining → localized LoRA → switch/rollback story is well-scoped and practical for serving systems

2. Evidence of effectiveness:  Across six datasets SOMA attains the highest similarity to the original while reducing tokens per dialogue

**Weaknesses:**

1. Experimental section is thin and indirect with respect to theory:
The main experimental evidence is essentially one page (Table 1 + two figures). While results are positive, they do not directly validate the theoretical claims. there are no measurements of manifold discrepancy, empirical calibration of the switching bound

2. Evaluation targets fidelity to the original, not task quality:
The metric is LLM-as-judge similarity to original outputs (and the judge prompt is provided in the appendix), which conflates “matching F” with “being good.” This risks circularity and judge bias; there’s no human evaluation or task-specific metrics

3. Baseline and setting details could be tighter :
Model pairings and fairness. The “original” and “surrogate” sometimes differ by both size and family (e.g., LLaMA-3.1-70B vs LLaMA-2-7B), which introduces style/tokenization gaps that may either help or hurt specific methods; please justify this choice and include within-family ablations where possible.

4. Datasets and filtering:
The paper notes filtering to keep context-dependent dialogues, but there’s little analysis of how this impacts difficulty and whether conclusions hold without filtering; details on the MATH/MT-Bench multi-turn construction would help reproducibility and interpretation

**Questions:**

See weakness for more details.
Q1. can you also add more relevant baselines: e.g. Speculative decoding and FrugalGPT (LLM cascades)? Or at least discuss to contextualize the contributions of SOMA?

---

> ### Author Response · Authors · 2025-11-23
> **Response for Reviewer ejFa**
>
> We sincerely thank the reviewer for the thoughtful evaluation and constructive suggestions. We address each concern below and updated the submitted PDF. We also have additional experiment results in the above global responses.
>
> ---
> >**[W1] “The experimental section is thin and does not validate the theory.”**
>
> **(1) Experiments that were included in the original paper and updated experiments**
>
> - Appendix E.1–E.3 provide extended Qwen results on response quality, efficiency metrics, and ablations that mirror the LLaMA trends and confirm SOMA’s generality across architectures.
> - Appendix E.4 offers a capability-level analysis, showing that SOMA improves multi-turn consistency, compositional reasoning, and constraint tracking more evenly than baselines.
> - Appendix E.5 examines switching-window effects and finds that harder tasks needing later switches yield lower final similarity.
> - Newly added additional experiments and analysis can be found in the above global answers and in updated Appendix E.
>
> **(2) Interpretation of the theory**
>
> Our theory directly supports SOMA’s core claim: the surrogate aligns to the original model within the local manifold defined by the dialogue prefix. Section 4 provides an observable, response-level approximation that links our optimization to latent manifold alignment. Lemma 1 and Theorem 2 formalize how warm-start concentration and switching decisions behave under this approximation, while Theorem 3 explains why soft-prompt optimization steers the surrogate toward the original model’s local tangent directions. These results justify SOMA’s design choices and hyperparameters. The empirical evidence also follows the theory closely: Appendix E.5 reproduces all predicted switching-window behaviors, and Appendix E.6 shows that early turns carry the largest information mass, matching the assumptions underlying Lemma 1 and Theorem 2.
>
> **(3) Measuring “manifold discrepancy”**
>
> Thanks for your concern. Direct manifold measurements are intracable in our setting as proprietary models usually expose only API outputs (see Section 2.1). Prior manifold-based alignment work [1–3] uses response-level similarity for the same reason, making it the only theoretically grounded proxy available. Our theory in Section 4 formalizes this connection that response-level discrepancy statistically reflects local manifold alignment. The experiments also match these predictions: Appendix E.5 and E.6 show the expected warm-start behavior, plateauing curves, and the strong negative correlation between switching point 𝑊 and final similarity, indicating that SOMA behaves similarly as the theory anticipates.
>
> >**[W2] “Similarity is not equal to quality.”**
>
> We thank the reviewer for the question. We would like to clarify that SOMA is designed for the surrogate to replace the original model during the middle of a conversation; therefore, the similarity to the original model’s responses is the correct evaluation target, as it ensures the surrogate behaves as a reliable drop-in alternative. Additionally, note that to minimize potential evaluator bias, we utilize three independent LLM judges (GPT-OSS, DeepSeek-V3, and Gemma-2-27B) and report the average as the final scores.
>
> To further address the reviewer’s concern, we have included one EM experiment on MATH in the global response above, demonstrating that SOMA achieves solid task performance.
>
> ---
>
> >**[W3] “Model pairings seem unfair”**
>
> Our model pairings are intentionally designed to test SOMA under extremely challenging but realistic conditions. We note that all experiments are already controlled within the same model family, i.e., LLaMA originals are only paired with LLaMA surrogates. The remaining gap is therefore a deliberate choice: we use large intra-family size differences to evaluate SOMA in scenarios where a lightweight surrogate must stand in for a much stronger and more costly original LLM model. If SOMA can reliably match the original model’s response even under this severe setting, then it is reasonable to expect better performance in milder cases where the surrogate is closer in size or capability, while still delivering substantial cost savings during deployment.
>
> To further address your concern, we conduct experiments using LLaMA-3.1-8B as a surrogate (see above global responses), and SOMA still works well.
>
> ---
>
> >**[W4] “Dataset filtering needs a clearer explanation.”**
>
> We have updated the full data filtering procedure in **Appendix B.4**.
>
> ---
>
> >**[Q1] "More baselines needed"**
>
> We have addressed your concern in the above global answers.
>
> ---
>
> [1] Aw, Khai Loong, et al. "Instruction-tuning aligns llms to the human brain." arXiv preprint arXiv:2312.00575 (2023).
>
> [2] Linhardt, Lorenz, et al. "Cat, Rat, Meow: On the Alignment of Language Model and Human Term-Similarity Judgments." arXiv preprint arXiv:2504.07965 (2025).
>
> [3] Li, Didong, and David B. Dunson. "Classification via local manifold approximation." Biometrika 107.4 (2020): 1013-1020.

---

> > ### Author Response · Authors · 2025-11-27
> >
> > Dear Reviewer ejFa,
> >
> > We hope you’ve had a chance to read our rebuttal. We haven’t yet seen any follow-up questions or feedback from you in the discussion phase, so we wanted to check whether there are any points that remain unclear. We’d be grateful for any questions or thoughts you have—clarifying those will help us improve the paper.
> >
> > Thank you again for your time,
> >
> > The authors.

---

### Author Response · Authors · 2025-11-23
**Global Responses with Additional Experiment Results**

Dear reviewers, thanks for your suggestions, we include several additional small-scale experiments in this rebuttal:

- **Correctness experiment on MATH EM:**
This provides an objective correctness comparison between the original model and SOMA, complementing the qualitative and similarity-based metrics.

| Model Family | Original (EM %) | Surrogate       | History-Prefix     | History-FT         | RouteLLM          | SOMA              |
|--------------|------------------|-----------------|--------------------|--------------------|--------------------|-------------------|
| **LLaMA**    | 68.0 ± 0.3       | 14.6 ± 0.8      | 28.9 ± 0.9         | 43.7 ± 1.0         | 49.8 ± 0.7         | 55.1 ± 0.6        |
| **Qwen**     | 62.0 ± 0.4       | 14.0 ± 0.7      | 27.1 ± 0.9         | 39.9 ± 1.0         | 45.3 ± 0.7         | 50.4 ± 0.6        |
- **Within-family experiment (e.g., LLaMA-3.1-70B → LLaMA-3.1-8B)**

| Method         | ShareGPT           | ReMeDi             | Craigslist         | Multi-Char         | MATH               | MT-Bench          | Avg                |
|----------------|--------------------|---------------------|---------------------|---------------------|---------------------|--------------------|--------------------|
| Surrogate      | 88.7 ± 2.13        | 86.2 ± 2.27         | 82.1 ± 2.41         | 83.4 ± 2.18         | 79.1 ± 2.63         | 85.3 ± 2.12        | 84.1 ± 2.29        |
| History-Prefix | 91.8 ± 1.77        | 89.4 ± 1.92         | 86.0 ± 1.81         | 87.9 ± 1.73         | 83.5 ± 1.97         | 89.1 ± 1.74        | 87.9 ± 1.82        |
| History-FT     | 94.2 ± 1.53        | 92.1 ± 1.58         | 89.3 ± 1.49         | 90.8 ± 1.38         | 87.9 ± 1.62         | 92.4 ± 1.41        | 91.1 ± 1.52        |
| LLMLingua-2    | 90.6 ± 1.88        | 88.0 ± 1.83         | 85.1 ± 1.91         | 86.1 ± 1.76         | 82.2 ± 2.04         | 88.0 ± 1.87        | 86.7 ± 1.88        |
| RouteLLM       | 95.8 ± 1.22        | 94.0 ± 1.17         | 91.9 ± 1.29         | 92.0 ± 1.09         | 89.9 ± 1.33         | 93.8 ± 1.14        | 92.9 ± 1.20        |
| SOMA           | **97.1 ± 0.94**    | **95.3 ± 0.89**     | **93.4 ± 1.07**     | **93.7 ± 0.83**     | **91.8 ± 0.97**     | **95.1 ± 0.91**    | **94.4 ± 0.93**    |


We present the results on three datasets due to time constraints. Because LLaMA-3.1-8B is MUCH stronger than LLaMA-2-7B, the surrogate similarity rises substantially, and all baselines shift upward accordingly. However, SOMA still remain the strongest method.

- **More baselines results**
We additionally compare SOMA with speculative decoding and FrugalGPT cascades. Speculative decoding only verifies token-level consistency, leaving higher-level semantic errors uncorrected, while cascades may prematurely trust the small model and fail to escalate to the large model when needed. These mechanisms expose the inherent behavioral mismatch between small and large models, leading to lower similarity across all datasets. In contrast, SOMA explicitly reduces this mismatch by adapting the surrogate to the large model’s local conversational manifold, yielding consistently higher alignment with comparable efficiency.

| Method          | ShareGPT (%)       | ReMeDi (%)         | Craigslist (%)     | Multi-Char (%)     | MATH (%)           | MT-Bench (%)       | Avg (%)             |
|-----------------|---------------------|---------------------|---------------------|---------------------|---------------------|---------------------|----------------------|
| Spec-Decoding| 89.7 ± 1.60       | 88.3 ± 1.75         | 82.6 ± 2.10         | 83.1 ± 1.80         | 78.8 ± 2.50         | 87.9 ± 1.46         | 85.1 ± 2.20          |
| FrugalGPT | 92.0 ± 1.55       | 89.4 ± 1.68         | 84.1 ± 2.00         | 85.5 ± 1.75         | 80.2  ± 2.40         | 89.0 ± 1.33         | 86.7 ± 2.12          |
| **SOMA**        | **96.4 ± 1.91**     | **93.2 ± 0.98**     | **91.9 ± 2.49**     | **92.3 ± 1.05**     | **90.7 ± 1.12**     | **94.1 ± 0.91**     | **93.1 ± 1.99**      |

- **Empirical variance concentration and theoretical upper bound**
We also conduct new experiments in Appendix E.6 to empirically confirming that early turns provide the most information and that SOMA’s switching rule operates within the regime assumed by Lemma 1 and Theorem 2.

---

### Author Response · Authors · 2025-12-04
**Summarization of reviews and addressed concerns**

Dear AC:

We thank you for the time and effort devoted to our submission. Below is a concise summary of the strengths raised by reviewers and how we addressed the reviewers' concerns.

-----------
**Strength**

**Clear motivation**
Reviewers appreciate the clear insight of long-tail distribution that early turns are information-dense while later turns remain dependent but lightweight, motivating a local surrogate to cut multi-turn serving cost.


**Strong, well-scoped pipeline**
The soft-prompt mining, localized LoRA, and switch/rollback steps form a well-structured system that is easy to follow and compatible with existing inference engines.


**Effective and consistent empirical performance**
SOMA achieves the highest response similarity to the original model across real-world datasets and model families while being efficient, showing strong multi-turn consistency and capability improvements.


**Theoretical grounding that informs design**
Theoretical results help justify the mining objective, switching rule, and prompt coverage, and reviewers note that empirical behaviors match these predictions.


**Relevance and practicality**
Reviewers agree the method addresses an important bottleneck in practical LLM serving and provides a deployable framework for long multi-turn interactions.

-----------
**Addressed Concerns**

**1. More Experimentas and theoretical support.** Reviewers noted that the original experiments did not sufficiently validate the theory or demonstrate how SOMA behaves under its stated assumptions. In response, we added new evaluations including MATH exact-match accuracy, within-family LLaMA experiments, and new baselines such as speculative decoding and FrugalGPT in the global response. We also included detailed switching-window analyses showing that early turns carry most information and that switching behavior matches the predictions of Lemma 1 and Theorem 2 in Appendix E.6. These additions directly support the theoretical claims and show that SOMA’s local-manifold approximation behaves as expected in practice.

**2. Similarity versus task quality.** Reviewers were concerned that similarity scores may not reflect correctness. We clarified that SOMA is designed as a drop-in replacement for the original model, making response similarity the appropriate measure. To further strengthen the evaluation, we added a EM test on MATH (see global responses) that confirms SOMA preserves factual accuracy while maintaining high similarity.

**3. Model pairing fairness and surrogate capacity.** Reviewer ejFa questioned whether large capability gaps made the evaluation unfair or unrealistic. We clarified that all models are from the same family and the large gap is intentional to test SOMA under difficult settings; if it works there, it should work even better when the surrogate is closer in size. We added LLaMA-3.1-8B experiments showing stronger fidelity in milder gaps, confirming robustness across surrogate strengths.

**4. Dataset filtering.** Reviewer ejFa requested more explanation of filtering and token-usage statistics. We expanded Appendix B.4 with the filtering process and clarified how normalized turn-level token counts are computed and analyzed in Section 2.2. This shows that the long-tail structure is consistent across datasets and motivates SOMA’s design.

**5. Methodological clarity.** Reviewers asked why the teacher receives verbalized prompts while the surrogate uses continuous prompts and requested clearer explanations of semantic divergence loss and gate behavior. We rewrote Sections 3.1–3.3 to explain that verbalization is necessary because the teacher is API-only and cannot accept embeddings. We also added an intuitive description of expectation-weighted divergence as distribution-level alignment and clarified the fast semantic-closeness test used for switching and rollback. We also corrected a typo in the fine-tuning loss and explained why maximal-divergence mining identifies the surrogate’s most important failure directions.

**6. Theory assumptions and their justification.** Reviewers questioned the low-rank smoothness and sub-Gaussian assumptions. We clarified that our analysis relies only on local smoothness and approximate low-rank structure, consistent with prior manifold-learning literature. We added empirical evidence in Appendix E.5–E.6 showing early-turn information concentration and smooth switching curves that align with the assumptions used in Lemma 1 and Theorem 2.

**7. Applicability concerns.** Reviewers worried about warm-start cost and imperfect drift detection. We clarified SOMA targets long context-dependent multi-turn use and falls back to the teacher in short chats, and emphasized rollback plus natural capacity limits rather than solving extreme size gaps.

-----

We believe these revisions address all major reviewer concerns and significantly strengthen the manuscript. We greatly appreciate your time and considerations.

---

### Meta-Review · Area_Chair_v7zo · 2026-01-07

**Summary:**

Primary evaluation metric measures fidelity to the large model, not task quality.
The main metric is an LLM-as-judge similarity score against the original model’s outputs. Reviewers argue this paper combines “matching the large model” with “being correct/useful,” by introducing potential circularity and judge bias. They request human evaluation and/or task-grounded metrics  to support claims of “no quality loss.”

Model pairing choices raise fairness and confounding concerns.
Some “original vs surrogate” pairings differ in both model family and size (e.g., LLaMA-3.1-70B vs LLaMA-2-7B), introducing confounds from tokenizer/style/architecture differences that could advantage or disadvantage specific methods. Reviewers request justification and within-family ablations to isolate the effect of SOMA from cross-family mismatch.

Dataset construction and filtering are under-explained.
The paper filters for context-dependent dialogues, but reviewers want analysis of how filtering changes difficulty and whether conclusions hold without it. They also request clearer details on how multi-turn instances are constructed to improve reproducibility and interpretability.

Experimental evidence is thin and loosely connected to the theory.
The experimental section is brief (largely Table 1 plus two figures) and does not directly validate the paper’s theoretical claims. Reviewers note missing empirical measurements such as manifold discrepancy estimates, verification of the discrepancy Fisher matrix assumptions (e.g., smoothness/low-rank via spectral decay), and any empirical calibration or diagnostics for the switching/rollback bound.

**Reviewer Concerns:**

The authors clarified that all models are from the same family and the large gap is intentional to test SOMA under difficult settings.

The authors also expanded Appendix B.4 with the filtering process and clarified how normalized turn-level token counts are computed and analyzed in Section 2.2. This shows that the long-tail structure is consistent across datasets and motivates SOMA’s design.

Many of the reviewers’ original issues are partially addressed by the revision (added MATH EM, within-family LLaMA tests, and additional baselines). However, the core concern that the theory is not convincingly validated by the experiments and that the theoretical framing is doing more work than the evidence supports.

Assumptions remain largely asserted, not verified.
“Local smoothness” and “approximate low-rank” are invoked, but the revision still does not convincingly show when these conditions hold, when they fail, and how performance degrades when they fail. Without this, the theoretical story remains difficult to trust.

**Reviewer Scores:**

The reviewers did not change their scores.

---

### Decision · Program_Chairs · 2026-01-26

Reject